# NFT: Bridging Supervised Learning and Reinforcement Learning in Math Reasoning

Huayu Chen[1,2]    Kaiwen Zheng[1,2]    Qinsheng Zhang[2]    Ganqu Cui[1]

Yin Cui[2]    Haotian Ye[2,3]    Tsung-Yi Lin[2]    Ming-Yu Liu[2]    Jun Zhu[1*]    Haoxiang Wang[2]

[1]Dept. of Comp. Sci. & Tech., BNRist Center, THU-Bosch ML Center, AI Institute, Tsinghua
[2]NVIDIA    [3]Stanford

**https://research.nvidia.com/labs/dir/Negative-aware-Fine-Tuning**

## Abstract

Reinforcement Learning (RL) has played a central role in the recent surge of LLMs' math abilities by enabling verification-driven training through binary verifier signals. In contrast, Supervised Learning (SL) is rarely considered for such verification-driven training, largely due to its heavy reliance on reference answers and inability to reflect on mistakes. In this work, we challenge the prevailing notion that self-improvement is exclusive to RL and propose Negative-aware Fine-Tuning (NFT) — a supervised approach that enables LLMs to reflect on their failures and improve autonomously with no external teachers. In online training, instead of throwing away self-generated negative answers, NFT constructs an *implicit* negative policy to model them. This implicit policy is parameterized with the same positive LLM we target to optimize on positive data, enabling direct policy optimization on all LLMs' generations. We conduct experiments on 7B and 32B models in math reasoning tasks. Results consistently show that through the additional leverage of negative feedback, NFT significantly improves over SL baselines like rejection fine-tuning, matching, or even surpassing leading RL algorithms like GRPO and DAPO. Furthermore, we demonstrate that NFT and GRPO are actually equivalent in strict-on-policy training, even though they have entirely different theoretical foundations. Our experiments and theoretical findings bridge the gap between SL and RL methods in binary-feedback learning systems.

## 1 Introduction

The recent surge in math reasoning abilities of Large Language Models (LLMs) is largely driven by a fundamental shift in their learning paradigm, from imitation to self-improvement (DeepSeek-AI, 2025; OpenAI, 2024; Chen et al., 2025c). Instead of relying on reference answers supplied by human annotators or stronger models (Liu et al., 2024; OpenAI, 2023), the new paradigm requires only a question dataset with a binary verifier to judge the correctness of self-generated answers. By reflecting on their own generation, LLMs can improve autonomously. This approach not only eliminates the need for costly data annotation but also removes competence ceilings imposed by external teachers, offering a promising path toward general intelligence (DeepSeek-AI, 2025; NVIDIA, 2025).

Reinforcement Learning (RL) appears to be a natural fit for such verification-driven training. Specific algorithms like PPO (Schulman et al., 2017) and GRPO (Shao et al., 2024) are explicitly designed to maximize reward signals, which can conveniently take the form of a binary verifier outcome. In contrast, Supervised Learning (SL) is rarely considered for realizing such verification-driven training. A common view holds that SL is inherently designed to memorize the positive data, rendering it unsuitable for self-reflective learning from negative mistakes (Chu et al., 2025a).

*In this work, we challenge the prevailing notion that verification-driven training is exclusive to RL, and demonstrate it can be similarly achieved within the supervised learning paradigm.*

---

*Corresponding author.

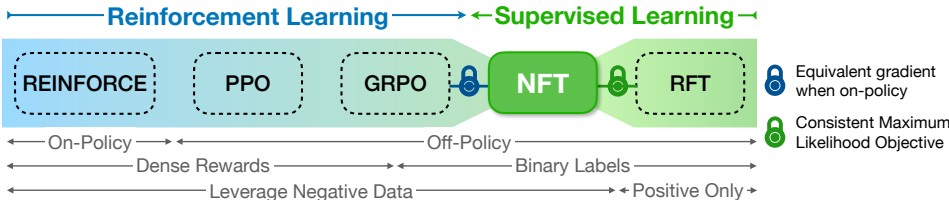

Figure 1: A spectrum of online algorithms for LLM fine-tuning. NFT bridges reinforcement learning and supervised learning methods through the leverage of negative feedback via supervision.

We start with a simple SL baseline: Rejection Fine-Tuning (RFT) (Yuan et al., 2023b; Dong et al., 2023). At each iteration, an LLM generates answers to questions. A verifier helps reject all negative answers. The remaining positive ones are compiled into a dataset to fine-tune the LLM itself in a supervised manner. RFT has been demonstrated effective (Bai et al., 2022; Yuan et al., 2023a; Song et al., 2023; Touvron et al., 2023; Xiong et al., 2025). However, it prevents any learning from negative feedback. LLMs are encouraged to reinforce what they already perform well, rather than reflect on their mistakes — an ability we believe critical for achieving general intelligence.

To overcome this limitation, we propose Negative-aware Fine-Tuning (NFT), an online learning algorithm that enables LLMs to learn from their negative generations (Sec. 3). Like RFT, NFT fine-tunes a positive LLM on positive answers via supervision. Crucially, instead of throwing away negative answers, NFT also constructs an *implicit* negative policy to model them. This implicit policy is parameterized with the same positive LLM we target to optimize on positive data, enabling direct policy optimization on all LLMs' generations (Figure 2). NFT has minimal memory overhead, as only a single model is maintained throughout training.

To understand the connection between NFT and RL approaches, we conduct an in-depth comparison between NFT and GRPO (Sec. 4). Surprisingly, we find the two methods are actually equivalent in *strict-on-policy* training, despite that they originate from entirely different theoretical frameworks (Figure 1). Notably, the "advantage normalization" characteristic of GRPO is already implicitly reflected in NFT's loss function. Their main difference arises in *off-policy* settings, regarding different strategies for clipping model gradients when the learned policy deviates from the old policy. These observations suggest a strong connection between SL and RL in binary-feedback learning systems.

We evaluate NFT on 7B and 32B Qwen models and report two key findings: 1. Supervised Learning alone can significantly enhance LLMs' math reasoning with no external teachers. NFT matches or even surpasses state-of-the-art RL algorithms like GRPO (Shao et al., 2024) and DAPO (Yu et al., 2025). 2. The performance gap between RFT and RL in online training largely stems from SL's past inability to leverage negative feedback, rather than any inherent superiority of RL. Through the additional leverage of negative data, NFT substantially mitigates this gap.

## 2 BACKGROUND

### 2.1 MAXIMUM LIKELIHOOD VS. POLICY GRADIENT

**Supervised Learning** essentially aims to learn a model $\pi_\theta(\boldsymbol{a}|\boldsymbol{q})$ to fit the data distribution $\pi(\boldsymbol{a}|\boldsymbol{q})$. This can be realized by employing the maximum-likelihood objective:

$$\max_\theta \mathbb{E}_{\boldsymbol{a}\sim\pi(\boldsymbol{a}|\boldsymbol{q})} \log \pi_\theta(\boldsymbol{a}|\boldsymbol{q}) \Leftrightarrow \min_\theta D_{\mathrm{KL}}\left[\pi(\boldsymbol{a}|\boldsymbol{q})\|\pi_\theta(\boldsymbol{a}|\boldsymbol{q})\right].$$

In LLM fine-tuning, $\boldsymbol{q}$ usually means the question prompt, and $\boldsymbol{a}$ the answer. To perform Maximum-likelihood training, we need a dataset $\mathcal{D} = \{\boldsymbol{q}, \boldsymbol{a} \sim \pi(\boldsymbol{a}|\boldsymbol{q})\}$ to draw training samples from.

**Reinforcement Learning**, by contrast, maximizes pre-defined reward $r(\boldsymbol{q}, \boldsymbol{a})$ for $\boldsymbol{a} \sim \pi_\theta(\boldsymbol{a}|\boldsymbol{q})$:

$$\max_\theta J(\theta) := \mathbb{E}_{\boldsymbol{a}\sim\pi_\theta(\boldsymbol{a}|\boldsymbol{q})} r(\boldsymbol{q}, \boldsymbol{a}).$$

Directly back-propagating through $J(\theta)$ is non-trivial, as $r(\cdot)$ can be an arbitrary scalar function whose gradient is unknown. Luckily, $\nabla_\theta J(\theta)$ can be estimated, making policy optimization feasible.

$$\nabla_\theta J(\theta) = \mathbb{E}_{\boldsymbol{a}\sim\pi_\theta(\boldsymbol{a}|\boldsymbol{q})} \nabla_\theta \left[r(\boldsymbol{q}, \boldsymbol{a}) \log \pi_\theta(\boldsymbol{a}|\boldsymbol{q})\right]. \tag{1}$$

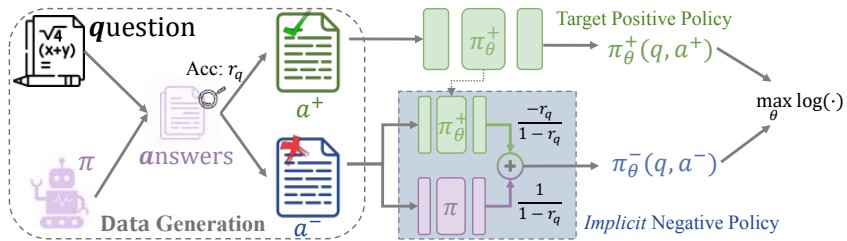

Figure 2: Illustration of the NFT algorithm. **Data Collection:** An LLM $\pi$ generates answers to a set of math questions. Generation results are split into two sub-datasets based on their answer correctness. **Policy Optimization:** By constructing an *implicit* policy for modeling negative data, NFT enables direct policy optimization on both positive and negative answers via maximum-likelihood.

Eq. 1 is known as the Policy Gradient (PG) algorithm (Sutton et al., 1999). In sequential decision-making problems such as language reasoning, $\boldsymbol{a}$ can be interpreted as the token decision for each step $t$, and $r(\boldsymbol{q}, \boldsymbol{a})$ can be replaced with advantage functions $\hat{A}(\boldsymbol{q}, \boldsymbol{a})$ (Schulman et al., 2015).

## 2.2 MATH REASONING RL: FROM POLICY GRADIENT TO GRPO

Eq. 1 requires training to be *on-policy*, where $\pi_\theta$ can only be updated a single time after data collection. To break this limitation, importance sampling can be applied (Schulman et al., 2017). Suppose the policy for collecting the RL dataset is denoted as $\pi_{\text{old}}$, we have

$$\nabla_\theta J(\theta) = \mathbb{E}_{\boldsymbol{a}\sim\pi_{\text{old}}(\boldsymbol{a}|\boldsymbol{q})} \left[ \frac{\pi_\theta(\boldsymbol{a}|\boldsymbol{q})}{\pi_{\text{old}}(\boldsymbol{a}|\boldsymbol{q})} r(\boldsymbol{q}, \boldsymbol{a}) \nabla_\theta \log \pi_\theta(\boldsymbol{a}|\boldsymbol{q}) \right] = \mathbb{E}_{\boldsymbol{a}\sim\pi_{\text{old}}(\boldsymbol{a}|\boldsymbol{q})} \left[ r(\boldsymbol{q}, \boldsymbol{a}) \nabla_\theta R_\theta(\boldsymbol{q}, \boldsymbol{a}) \right],$$
(2)

where $R_\theta(\boldsymbol{q}, \boldsymbol{a}) := \frac{\pi_\theta(\boldsymbol{a}|\boldsymbol{q})}{\pi_{\text{old}}(\boldsymbol{a}|\boldsymbol{q})}$ is the likelihood Ratio between two policies.

In math reasoning tasks, PPO (Schulman et al., 2017) and subsequent GRPO (Shao et al., 2024) algorithms further apply gradient clipping to constrain the distance between $\pi_\theta$ and $\pi_{\text{old}}$:

$$\mathcal{L}^{\text{GRPO}}(\theta) = -\sum_{\boldsymbol{q}, \boldsymbol{a}\sim\pi_{\text{old}}} \sum_t \min\left[ R_\theta^t(\boldsymbol{q}, \boldsymbol{a})\hat{A}_{\boldsymbol{q},\boldsymbol{a}}, \text{clip}(R_\theta^t(\boldsymbol{q}, \boldsymbol{a}), 1-\epsilon', 1+\epsilon')\hat{A}_{\boldsymbol{q},\boldsymbol{a}} \right],$$
(3)

where $R_\theta^t(\boldsymbol{q}, \boldsymbol{a}) := \frac{\pi_\theta(\boldsymbol{a}_t|\boldsymbol{q}, \boldsymbol{a}_{<t})}{\pi_{\text{old}}(\boldsymbol{a}_t|\boldsymbol{q}, \boldsymbol{a}_{<t})}$, and $\hat{A}_{\boldsymbol{q},\boldsymbol{a}}$ is the estimated advantage value. Note that we have dropped some auxiliary loss terms, such as KL and entropy regularization, as they have been pointed out to be unnecessary by recent studies like DAPO (Yu et al., 2025).

GRPO proposes an efficient way to estimate $\hat{A}_{\boldsymbol{q},\boldsymbol{a}}$. Collect $K$ answers $\boldsymbol{a}^{1:K}$ and their binary reward $r^{1:K} \in \{0, 1\}$ for each question. The advantage is defined to be the normalized reward:

$$\hat{A}_{\boldsymbol{q},\boldsymbol{a}} := \left[ r(\boldsymbol{q}, \boldsymbol{a}) - \text{mean}\{r^{1:K}\} \right] / \text{std}\{r^{1:K}\}.$$
(4)

Later studies (Liu et al., 2025b) suggest removing the std term from Eq. 4.

## 3 METHOD

### 3.1 PROBLEM SETUP

**Dataset.** Given a set of $N$ math questions $\{\boldsymbol{q}^{1:N}\}$, a pretrained LLM $\pi_{\text{old}}(\boldsymbol{a}|\boldsymbol{q})$, and a verifier for judging the correctness. In every iteration, we generate a dataset $\mathcal{D} = \{\boldsymbol{q}, \boldsymbol{a}^{1:K} \sim \pi, r^{1:K}\}^{1:N}$, where $r \in \{0, 1\}$ is the correctness label, and $K$ the number of answers collected for each question.

Dataset $\mathcal{D} \sim \pi$ can be split into two subsets: $\mathcal{D}^+$ and $\mathcal{D}^-$. $\mathcal{D}^+$ contains all positive answers, and $\mathcal{D}^-$ contains the rest negative ones. We denote the underlying answer distribution of $\mathcal{D}^+$ as $\pi^+(\cdot|\boldsymbol{q})$.

**Learning Target.** We want to optimize the old policy $\pi$ into a new policy $\pi_\theta^+ \approx \pi^+$. The target $\pi^+(\boldsymbol{a}|\boldsymbol{q})$ can be formalized using Bayes' Rule:

$$\pi^+(\boldsymbol{a}|\boldsymbol{q}) := \pi(\boldsymbol{a}|\boldsymbol{q}, r{=}1) = \frac{\pi_{\text{old}}(\boldsymbol{a}|\boldsymbol{q})p(r{=}1|\boldsymbol{q}, \boldsymbol{a})}{\sum_A \pi_{\text{old}}(\boldsymbol{a}|\boldsymbol{q})p(r{=}1|\boldsymbol{q}, \boldsymbol{a})},$$
(5)

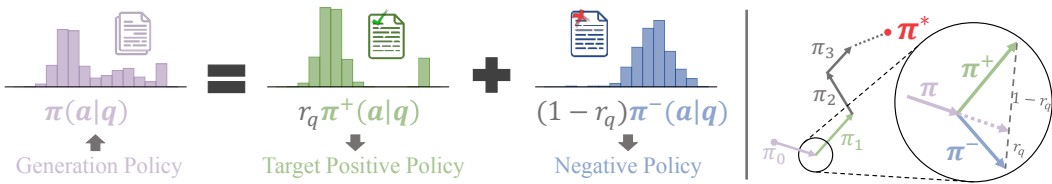

Figure 3: **Left:** Policy Splitting. The generation policy can be split into a positive policy and a negative policy, and re-expressed as their linear combination. **Right:** Policy Improvement. By iteratively optimizing towards its positive split, an LLM policy $\pi_0$ can improve continuously.

where $A$ means all possible language space for $\boldsymbol{a}$.

**Discussion.** An obvious solution for learning $\pi^+$ is to fine-tune solely on correct answers ($\mathcal{D}^+$) and discard $\mathcal{D}^-$ totally (RFT) (Dong et al., 2023; Xiong et al., 2025). However, this approach prevents the model from any learning on its negative feedback ($\mathcal{D}^-$). We posit that the ability to reflect on one's own failures is not merely desirable, but central to general intelligence, marking a shift from pure imitation to self-reflective learning. Though traditionally viewed as a distinctive strength of RL (Chu et al., 2025a; Yue et al., 2025), we ask: *Can self-reflective improvement be similarly achieved within the SL paradigm?*

## 3.2 DIRECT OPTIMIZATION OF LANGUAGE MODELS WITH NEGATIVE ANSWERS

In this section, we discuss how to leverage negative data $\mathcal{D}^-$ to directly optimize $\pi_\theta^+$. Despite seemingly impossible at first thought, we find the target policy $\pi^+$ and the negative policy $\pi^-$ are tightly coupled, making feasible training $\pi_\theta^+$ directly from $\mathcal{D}^-$.

First, we formalize the definition of the negative policy $\pi^-$ similar to Eq. 5

$$\pi^-(\boldsymbol{a}|\boldsymbol{q}) := \pi(\boldsymbol{a}|\boldsymbol{q}, r{=}0) = \frac{\pi_{\text{old}}(\boldsymbol{a}|\boldsymbol{q})[1 - p(r{=}1|\boldsymbol{q},\boldsymbol{a})]}{\sum_A \pi_{\text{old}}(\boldsymbol{a}|\boldsymbol{q})[1 - p(r{=}1|\boldsymbol{q},\boldsymbol{a})]}. \tag{6}$$

Combining Eq. 5 and Eq. 6, we make a key observation that

$$r_{\boldsymbol{q}}\pi^+(\boldsymbol{a}|\boldsymbol{q}) + [1 - r_{\boldsymbol{q}}]\pi^-(\boldsymbol{a}|\boldsymbol{q}) = \pi_{\text{old}}(\boldsymbol{a}|\boldsymbol{q}), \tag{7}$$

where $r_{\boldsymbol{q}} := \sum_A \pi_{\text{old}}(\boldsymbol{a}|\boldsymbol{q})p(r{=}1|\boldsymbol{q},\boldsymbol{a}) = p(r{=}1|\boldsymbol{q})$ is the correctness rate of LLM $\pi_{\text{old}}$ over a question $\boldsymbol{q}$. In practice, $r_{\boldsymbol{q}} \approx \text{mean}\{r^{1:K}\}$ can be estimated using the $K$ rewards in dataset $\mathcal{D}$.

**Implicit negative policy.** Eq. 7 reveals a tight coupling between $\pi^+$ and $\pi^-$ (Figure 3). Given that we already have $\pi$ as the pretrained LLM and $r_{\boldsymbol{q}}$ is estimable, learning $\pi^-$ from negative data should, in principle, shape the target policy $\pi_\theta^+$, in a manner analogous to SL on positive data.

To realize this idea, we construct an *implicit* negative policy, denoted $\pi_\theta^-$, by re-parameterizing the target policy $\pi_\theta^+$ using the relationship in Eq. 7:

$$\pi_\theta^-(\boldsymbol{a}|\boldsymbol{q}) := \frac{\pi_{\text{old}}(\boldsymbol{a}|\boldsymbol{q}) - r_{\boldsymbol{q}}\pi_\theta^+(\boldsymbol{a}|\boldsymbol{q})}{1 - r_{\boldsymbol{q}}}.$$

Thus, training $\pi_\theta^-$ on negative answers directly leads to optimizing the underlying positive LLM $\pi_\theta^+$ (Figure 2). We have the following guarantee:

**Theorem 3.1 (Policy Optimization with Negative Answers).** *Consider the maximum-likelihood objective for training the implicit negative policy $\pi_\theta^-$:*

$$\max_\theta \mathbb{E}_{p(\boldsymbol{q})\pi^-(\boldsymbol{a}|\boldsymbol{q})}\left[\log \pi_\theta^-(\boldsymbol{a}|\boldsymbol{q})\right] \Leftrightarrow \min_\theta \left[-\mathbb{E}_{(\boldsymbol{q},\boldsymbol{a})\sim\mathcal{D}^-} \log \frac{\pi_{old}(\boldsymbol{a}|\boldsymbol{q}) - r_{\boldsymbol{q}}\pi_\theta^+(\boldsymbol{a}|\boldsymbol{q})}{1 - r_{\boldsymbol{q}}}\right] \tag{8}$$

*Assuming unlimited data and model capacity, the optimal solution for solving Eq. 8 is*

$$\forall \boldsymbol{q}, \boldsymbol{a}: \quad \pi_{\theta*}^+(\boldsymbol{a}|\boldsymbol{q}) = \pi^+(\boldsymbol{a}|\boldsymbol{q})$$

---

**Algorithm 1** Negative-aware Fine-Tuning (NFT)

---

1: **Input:** Language model $\pi$, prompt set $\boldsymbol{q}^{1:N}$, verifier $r(\cdot)$.
2: **Def** max_v$(\boldsymbol{x}, \epsilon)$**:**                                             *//Straight-through Max Operator*
3:     **Return** stop_gradient$[\max(\boldsymbol{x}, \epsilon) - \boldsymbol{x}] + \boldsymbol{x}$           *//Clip Value while Keeping Gradient*
4: **for** each iteration **do**
5:     **for** each sampled prompt $\boldsymbol{q}$ **do**
6:         Generate $K$ answers $\boldsymbol{a}^{1:K}$ and verify their correctness $r^{1:K}$        *// Data Collection*
7:         Calculate correctness rate $\hat{r}_{\boldsymbol{q}} = \text{mean}\{r^{1:K}\}$ and token-level likelihood $\{\pi_{\text{old}}(\boldsymbol{a}_t | \boldsymbol{q}, \boldsymbol{a}_{<t})^{1:|\boldsymbol{a}|}\}^{1:K}$
8:         $\mathcal{D} \leftarrow \{\boldsymbol{q}, \hat{r}_{\boldsymbol{q}}, \boldsymbol{a}^{1:K}, r^{1:K}, \pi_t^{1:K}\}$ **If** $0 < r_{\boldsymbol{q}} < 1$          *// Prompt Filtering*
9:     **end for**
10:    Initialize $\pi_\theta^+ \leftarrow \pi$
11:    **for** each mini batch $\{\boldsymbol{q}, \boldsymbol{a}, r, \hat{r}_{\boldsymbol{q}}, \pi_t\}$ in $\mathcal{D}$ **do**
12:        $R_\theta^t(\boldsymbol{q}, \boldsymbol{a}) = \frac{\pi_\theta^+(\boldsymbol{a}_t | \boldsymbol{q}, \boldsymbol{a}_{<t})}{\pi_{\text{old}}(\boldsymbol{a}_t | \boldsymbol{q}, \boldsymbol{a}_{<t})}, \forall t$           *// Positive Likelihood Ratio*
13:        **If** r = 0:
14:           $R_\theta^t(\boldsymbol{q}, \boldsymbol{a}) = (1 - \hat{r}_{\boldsymbol{q}} R_\theta^t(\boldsymbol{q}, \boldsymbol{a})) / (1 - \hat{r}_{\boldsymbol{q}})$       *// Implicit Negative Likelihood Ratio*
15:           $R_\theta^t(\boldsymbol{q}, \boldsymbol{a}) = \text{max\_v}[R_\theta^t(\boldsymbol{q}, \boldsymbol{a}), \epsilon]$          *// Clip Negative Likelihood Ratio*
16:        $\theta \leftarrow \theta + \lambda \nabla_\theta \sum_t \log R_\theta^t(\boldsymbol{q}, \boldsymbol{a})$   (Eq. 10)      *// Maximum Likelihood Training*
17:    **end for**
18:    $\pi \leftarrow \pi_\theta^+$, start the next iteration
19: **end for**

---

Proof in Appendix A. Theorem 3.1 demonstrates the feasibility of policy optimization on negative data only. To further utilize positive data, we additionally conduct supervised training on $\mathcal{D}^+$:

$$\mathcal{L}_{(\boldsymbol{a}, \boldsymbol{q}, r) \sim \mathcal{D}}^{\text{NFT}}(\theta) = r \left[ -\log \frac{\pi_\theta^+(\boldsymbol{a} | \boldsymbol{q})}{\pi_{\text{old}}(\boldsymbol{a} | \boldsymbol{q})} \right] + (1 - r) \left[ -\log \frac{1 - r_{\boldsymbol{q}} \frac{\pi_\theta^+(\boldsymbol{a} | \boldsymbol{q})}{\pi_{\text{old}}(\boldsymbol{a} | \boldsymbol{q})}}{1 - r_{\boldsymbol{q}}} \right] \tag{9}$$

Note that we subtract a baseline term $-\log \pi_{\text{old}}(\boldsymbol{a} | \boldsymbol{q})$ from the loss. Since this term is unrelated to $\theta$, it does not affect the loss gradient and thus the optimal solution. $\pi_{\text{old}}(\boldsymbol{a} | \boldsymbol{q})$ is the old data likelihood before optimizer update. At the start of training, we have $\pi_\theta^+ = \pi$ such that $\mathcal{L}_{(\boldsymbol{a}, \boldsymbol{q}, r)}^\theta = 0$.

We name our method Negative-aware Fine-Tuning (NFT) as it enables the additional leverage of negative data for policy optimization compared with RFT.

**Continuous Reward.** Though we mainly consider binary rewards for math reasoning, NFT is directly applicable to continuous rewards. Following Levine (2018), we can define optimality reward $r \in [0, 1]$, meaning the answer has a probability of $r$ for being positive, and $1 - r$ for being negative. The loss definition in Eq. 9 and its convergence property stay unchanged (Proof in Appendix B).

**Memory Efficiency.** NFT is memory-efficient. In practice, we keep only a single model copy in memory. The old policy likelihood $\pi_{\text{old}}(\boldsymbol{a} | \boldsymbol{q})$ can be pre-computed during data generation.

### 3.3 PRACTICAL ALGORITHM

We introduce several improvements over Eq. 9 and propose a practical objective of NFT:

$$\mathcal{L}_{\mathcal{D}}^{\text{NFT}}(\theta) = -\sum_{\boldsymbol{q}, \boldsymbol{a}, r} \omega(\boldsymbol{q}) \sum_t \left[ r \log R_\theta^t(\boldsymbol{q}, \boldsymbol{a}) + (1 - r) \log \text{max\_v}(\frac{1 - \hat{r}_{\boldsymbol{q}} R_\theta^t(\boldsymbol{q}, \boldsymbol{a})}{1 - \hat{r}_{\boldsymbol{q}}}, \epsilon) \right] \tag{10}$$

$$\text{where} \quad R_\theta^t(\boldsymbol{q}, \boldsymbol{a}) = \frac{\pi_\theta^+(\boldsymbol{a}_t | \boldsymbol{q}, \boldsymbol{a}_{<t})}{\pi_{\text{old}}(\boldsymbol{a}_t | \boldsymbol{q}, \boldsymbol{a}_{<t})}, \quad \text{and} \quad \hat{r}_{\boldsymbol{q}} = \frac{1}{K} \sum_{\boldsymbol{a} | \boldsymbol{q}} r(\boldsymbol{q}, \boldsymbol{a}).$$

Pseudo code is in Algorithm 1. Below, we explain our key design choices.

**Token-level loss.** Eq. 9 essentially deals with sequence data, where answer likelihood $\pi(\boldsymbol{a} | \boldsymbol{q}) = \prod_t \pi(\boldsymbol{a}_t | \boldsymbol{q}, \boldsymbol{a}_{<t})$ is heavily correlated with answer length. This introduces high variance in gradient estimation and causes numerical instabilities during training. Following Schulman et al. (2017); Yu et al. (2025), we view each token decision as an individual unit and sum up their loss in Eq. 10.

**Clipping negative likelihood ratio.** The negative loss in Eq. 10 involves a logarithm whose argument must remain positive, imposing $(1 - \hat{r}_q R_\theta^t)/(1 - \hat{r}_q) > 0$. When $R_\theta^t$ is unoptimized, this requirement may not be satisfied, potentially leading to training collapse. We therefore enforce a minimum value of $\epsilon > 0$ for the negative likelihood ratio. To preserve gradient flow after clipping, we further apply straight-through gradient (Bengio et al., 2013). Details are in Algorithm 1.

**Prompt weighting.** To focus training on more informative instances, we weight each prompt $q$ by $\omega(q)$, assigning higher importance to hard questions with low $r_q$. An ablation study is posted in Sec. 5.4. This design also helps align NFT with RL algorithms like GRPO (Sec. 4).

## 4 UNDERSTANDING THE GAP BETWEEN NFT AND GRPO

Despite originating from entirely different theoretical foundations, NFT and GRPO exhibit significant similarities. Notably, we find **GRPO and NFT are equivalent in on-policy training**. To understand this, we calculate and compare their loss gradients:

**Proposition 4.1** (**Algorithm Gradient Comparision**). *Suppose there are $\hat{r}_q K$ positive answers and $(1 - \hat{r}_q)K$ negative ones for a given question $q$*

*(a) GRPO Gradient:* *Consider only binary reward $\{0, 1\}$ in Eq. 3, GRPO loss gradient satisfies*

$$\nabla_\theta \mathcal{L}_\mathcal{D}^{GRPO}(\theta) = -\sum \left\{ r A_q^+ \cdot \mathcal{I}\left[R_\theta^t(q, a) < 1 + \epsilon'\right] + (1-r)A_q^- \cdot \mathcal{I}\left[R_\theta^t(q, a) > 1 - \epsilon'\right] \right\} \nabla_\theta R_\theta^t(q, a),$$

*where $A_q^+ = \sqrt{\frac{1-\hat{r}_q}{\hat{r}_q}}$ and $A_q^- = -\sqrt{\frac{\hat{r}_q}{1-\hat{r}_q}}$ are respectively normalized advantages for answers.*

*(b) NFT Gradient:* *Let $\omega(q) = \sqrt{(1 - \hat{r}_q)/\hat{r}_q}$, NFT loss gradient satisfies*

$$\nabla_\theta \mathcal{L}_\mathcal{D}^{NFT}(\theta) = -\sum \left\{ r A_q^+ \cdot \frac{1}{R_\theta^t(q, a)} + (1-r)A_q^- \cdot \max\left[\frac{1 - \hat{r}_q\, R_\theta^t(q, a)}{1 - \hat{r}_q}, \epsilon\right]^{-1} \right\} \nabla_\theta R_\theta^t(q, a).$$

Proofs are provided in Appendix A. Comparing $\nabla_\theta \mathcal{L}_\mathcal{D}^{\mathrm{GRPO}}(\theta)$ and $\nabla_\theta \mathcal{L}_\mathcal{D}^{\mathrm{NFT}}(\theta)$, the only difference between GRPO and NFT is their strategy for clipping model gradients when training data becomes *off-policy* (Figure 4). GRPO simply zeros out the gradient when the learned policy $\pi_\theta$ shifts far away from the old policy $\pi$, while NFT takes a "softer" decay schedule.

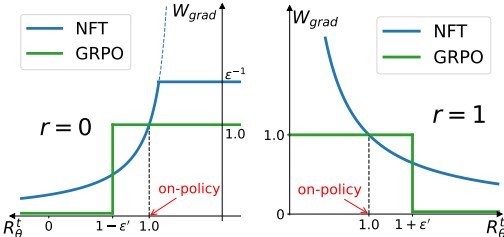

Figure 4: Gradient weight for NFT and GRPO.

Surprisingly, we find GRPO and NFT to be equivalent when training is totally on-policy, despite their distinctively different derivations:

**Proposition 4.2** (**On-policy Gradient Equivalence**). *Following the set up of Proposition 4.1 and let $\epsilon \leq 1$, GRPO and NFT loss gradient are equivalent in on-policy training:*

$$R_\theta^t(q, a) = 1 \quad \implies \quad \nabla_\theta \mathcal{L}_\mathcal{D}^{NFT}(\theta) = \nabla_\theta \mathcal{L}_\mathcal{D}^{GRPO}(\theta)$$

**Implicit group normalization.** Proposition 4.1 shows the "normalized advantage" term is implicitly present within NFT's loss function. This partially justifies the Group Normalization design choice for GRPO, which was initially introduced only as an empirical technique (Shao et al., 2024). In Appendix A, we further demonstrate that by adjusting $\omega(q) = 1 - \hat{r}_q$, NFT also aligns with Dr. GRPO (Liu et al., 2025b). Our findings suggest a strong connection between RL and SL frameworks.

## 5 EXPERIMENTS

We seek to answer the following questions through our experiments.

1. How does NFT perform in comparison with existing RL algorithms such as GRPO? (Sec. 5.2)
2. How does negative data contribute to NFT's performance gain? (Sec. 5.3)
3. Which empirical design choices are important to the effectiveness of NFT? (Sec. 5.4)

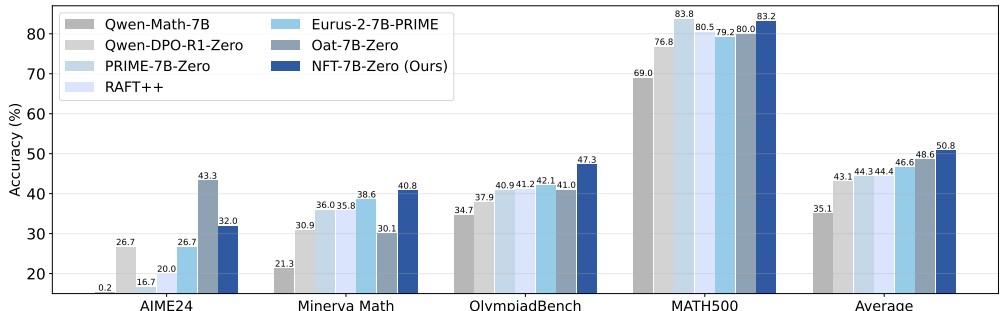

Figure 5: Comparison of the released NFT-7B with other zero-style math models of Qwen series.

Table 1: NFT performs competitively compared with other algorithms. We report `avg@32` for AIME24, AIME25, and AMC23 and `avg@1` for others. Numbers within 1 % of the max are bolded.

| Model | AIME24 | MATH500 | AIME25 | AMC23 | Olympiad | Minerva | Average |
|---|---|---|---|---|---|---|---|
| Qwen2.5-Math-7B | 13.3 | 69.0 | 5.5 | 45.8 | 34.7 | 21.3 | 31.6 |
| *Preference fine-tuning* | | | | | | | |
| + DPO | 29.8 | 79.8 | 13.8 | 83.2 | 48.0 | 39.0 | 48.9 |
| *Reinforcement fine-tuning* | | | | | | | |
| + GRPO | 30.2 | 80.4 | 17.1 | 79.5 | **51.8** | 38.2 | 49.5 |
| + Dr. GRPO | 31.8 | **83.4** | 15.7 | 80.2 | 49.6 | 38.2 | 49.8 |
| + DAPO | **33.1** | 81.6 | **18.7** | 85.0 | 49.9 | 39.3 | **51.2** |
| *Supervised fine-tuning* | | | | | | | |
| + RFT | **33.7** | 79.8 | 13.4 | 79.7 | 44.3 | 38.6 | 48.3 |
| + **NFT** | 32.0 | **83.2** | **18.3** | **88.5** | 47.3 | **40.8** | **51.7** |
| Qwen2.5-32B | 4.1 | 68.6 | 1.0 | 45.0 | 31.1 | 27.9 | 29.6 |
| + DAPO | **44.1** | **89.2** | **33.4** | 90.9 | **54.1** | 47.5 | **59.9** |
| + RFT | 29.9 | 86.2 | 19.1 | 92.4 | 45.3 | 44.1 | 52.8 |
| + **NFT** | 37.8 | **88.4** | 31.5 | **93.8** | **55.0** | **48.9** | 59.2 |

## 5.1 EXPERIMENT SETUPS

**Training.** We perform online fine-tuning on `Qwen2.5-Math-7B` and `Qwen2.5-32B` (Yang et al., 2024) to enhance their math abilities without relying on external teachers. The training dataset is the publicly available `DAPO-Math-17k` (Yu et al., 2025), which consists solely of math questions paired with ground-truth answers in integer form. During training, all models are fine-tuned for approximately 5,000 gradient steps with a batch size of 512. Generation temperature is 1.0.

**Evaluation.** We evaluate models on six validation benchmarks and report their average accuracy: `AIME 2024`, `AIME 2025`, `AMC 2023` (Li et al., 2024), `MATH500` (Hendrycks et al., 2021), `OlympiadBench` (He et al., 2024), and `Minerva Math` (Lewkowycz et al., 2022).

**Baseline methods.** We compare against a set of online fine-tuning algorithms, including Iterative DPO (Rafailov et al., 2023; Xiong et al., 2023; Guo et al., 2024), GRPO (Shao et al., 2024), Dr. GRPO (Liu et al., 2025b), DAPO (Yu et al., 2025), and RFT (Dong et al., 2023; Yuan et al., 2023b).

## 5.2 NFT PERFORMANCE EVALUATION

**Model comparison.** By applying NFT to Qwen2.5-Math-7B, we release `NFT-7B-Zero` (Figure 5). `NFT-7B-Zero` achieves competitive performance on all benchmarks compared to other zero-style 7B math models (Cui et al., 2025; Liu et al., 2025b; Xiong et al., 2025; 2023). This provides strong empirical evidence for the effectiveness of the NFT algorithm and demonstrates that SL alone can enable effective verification-driven training in math tasks.

**Algorithm comparison.** To isolate the contribution of the algorithm itself, we benchmarked various online algorithms using identical training data, infrastructure, and general hyperparameters (Table 1). Results show that NFT matches or even surpasses state-of-the-art methods such as DAPO. Fig-

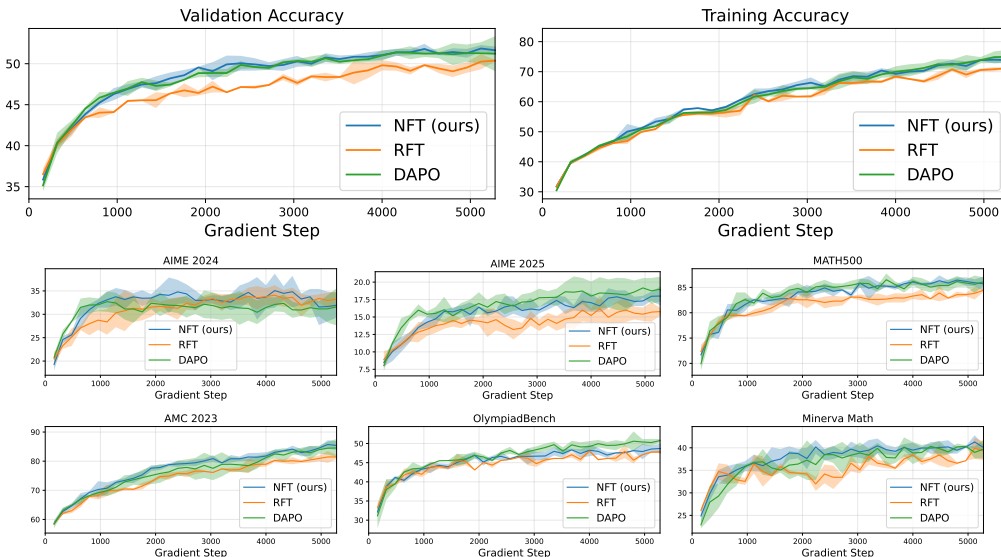

Figure 6: Training and validation accuracy curves for 7B experiments. We conducted 3-4 random and independent experiments for each algorithm and report their mean ± std results.

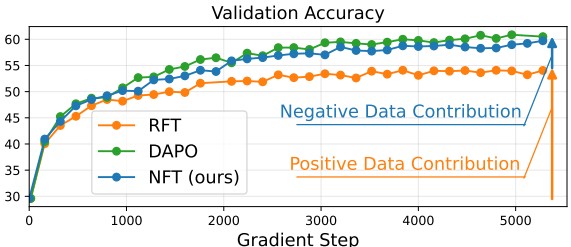

Figure 7: Average accuracy across 6 benchmarks for 32B experiments. More curves in Appendix D.

ure 6 and 11 present training curves across multiple runs. NFT exhibits convergence speed and final performance on par with DAPO, further supporting its stability.

## 5.3 BENEFITS OF NEGATIVE DATA

**Negative feedback enhances performance and exploration.** Table 1 shows that NFT consistently outperforms RFT by a clear margin, highlighting the benefit of incorporating negative feedback during training. Notably, we observe a clear divergence in training dynamics between RFT and NFT. Across both 7B and 32B settings, RFT tends to reduce entropy over time, whereas NFT and RL methods like DAPO encourage increasing entropy (Figure 8). This suggests more exploration (Yu et al., 2025), potentially hindering why NFT outperforms RFT.

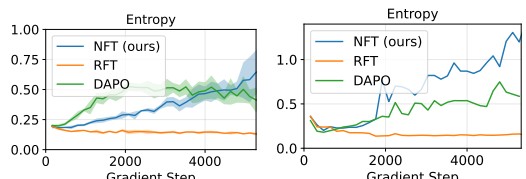

Figure 8: Entropy curves for 7B and 32B runs.

**Negative feedback becomes increasingly important in larger models.** The performance gap between RFT and NFT widens faster over training in 32B experiments compared with the trend in 7B (Figure 11). Similarly, DeepSeek-AI (2025) also notes RL offers greater benefits over SFT in larger models. A potential explanation could be the increasing importance of negative data: Larger models already memorize well enough, so the ability to reflect on mistakes becomes a new bottleneck.

**RFT remains a strong baseline.** Although surpassed by numerous algorithms, RFT still deserves attention due to its extreme simplicity. In 32B settings (Figure 11), learning from positive data (RFT) contributes to 80% of the total gain achieved by our best-performing model, while negative

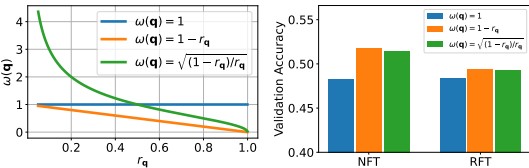

Figure 9: Effect of prompt weighting.

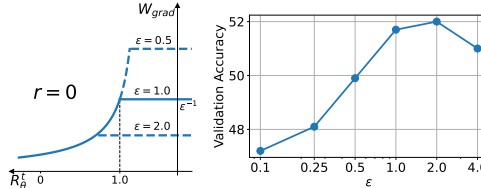

Figure 10: Effect of negative ratio clip value $\epsilon$.

data only accounts for the remaining 20%. These findings echo recent studies (Yue et al., 2025; Xiong et al., 2025; Liu et al., 2025a; Zhao et al., 2025; Wang et al., 2025), which suggest RL primarily amplifies existing capabilities in large models rather than fostering new skills. How to exploit negative feedback remains an open challenge with heavy potential.

### 5.4 INGREDIENTS BEHIND NFT'S EFFECTIVENESS

We discuss two empirical design choices that notably help NFT achieve strong performance.

**Prioritize harder questions.** We find that assigning higher weights to difficult questions with a low answer correctness rate $\hat{r}_q$ can enhance model performance. We achieve this mainly by selecting $\omega(\boldsymbol{q})$ in Eq. 10 from 3 choices: (1) $\omega(\boldsymbol{q}) = 1$. (2) $\omega(\boldsymbol{q}) = 1 - r_q$, which aligns NFT with Dr. GRPO in on-policy training (Sec. 4). (3) $\omega(\boldsymbol{q}) = \sqrt{(1 - r_q)/r_q}$, which aligns NFT with GRPO. Figure 9 visualizes different $\omega(\boldsymbol{q})$, and their effect for NFT and RFT. We find choices (2) and (3) perform similarly, both outperforming constant weighting choice (1).

**Avoid overpenalizing mistakes.** The clip value $\epsilon$ of NFT (Eq. 10) sets an upper bound on the penalty weight when the likelihood ratio $R_\theta^t$ for negative answers increases. When $\epsilon$ is small (e.g., near zero), the algorithm empirically assigns high penalties to rising likelihoods of incorrect answers (Figure 10). However, our experiments show that overly aggressive penalization with $\epsilon \to 0$ degrades overall performance. We thus adopt a default setting of $\epsilon = 1.0$.

## 6 RELATED WORKS

Reinforcement Learning with Verifiable Rewards (RLVR) has advanced the frontier of LLM reasoning (DeepSeek-AI, 2025; OpenAI, 2024; Team et al., 2025; Chen et al., 2025c). Compared with previous RL practices that rely on strong reward models (Wang et al., 2023b; Yuan et al., 2024; Zhang et al., 2024) to simulate human feedback (Ouyang et al., 2022; Christiano et al., 2017; Cui et al., 2025), RLVR turns to a ground truth verifier for providing reliable binary supervision (Lambert et al., 2024; Shao et al., 2024). Moreover, unlike preference-based learning algorithms such as DPO (Rafailov et al., 2023; Cai et al., 2023; Azar et al., 2024; Ethayarajh et al., 2024; Wang et al., 2023a; Chen et al., 2024; Hong et al., 2024; Xu et al., 2023), RLVR does not require paired preference data, rendering it more flexible and memory-efficient. Despite the demonstrated effectiveness of RL algorithms in verification-driven training (Li et al., 2023; Ahmadian et al., 2024; Hu, 2025; Yu et al., 2025; Chu et al., 2025b; Yuan et al., 2025; Yan et al., 2025), recent studies suggest that supervised learning (SL) may also suffice for achieving verification-driven training in LLMs (Dong et al., 2023). Our method further addresses SL's inability to incorporate negative feedback (Hua et al., 2024), bridging both the theoretical and the performance gap between the two fields.

A key design of NFT involves implicit policy modeling for direct policy optimization. This design, emphasizing direct optimization via implicitly defined models, shares conceptual similarities with some existing approaches. In preference-based training, DPO (Rafailov et al., 2023) introduces an implicit reward model parameterized by the policy network to allow optimizing policies directly. Recent visual modeling efforts also leverage implicit conditional or residual models parameterized by generation networks to avoid guided sampling (Chen et al., 2025b;a; Zheng et al., 2025).

## 7 CONCLUSION

In this work, we introduce Negative-aware Fine-Tuning (NFT), a supervised approach that enables LLMs to learn from their own negative generations. In online training, NFT substantially improves upon supervised learning baselines through the additional leverage of negative feedback, achieving performance comparable to leading RL algorithms like GRPO. Notably, we unveiled a theoretical equivalence between NFT and GRPO under strict-on-policy conditions. These findings bridge the conceptual and practical gap between SL and RL paradigms in verification-driven training.

### THE USE OF LARGE LANGUAGE MODELS (LLMS)

We used large language models (LLMs) solely as a writing assistant for language polishing and improving clarity of presentation. The LLMs were not involved in research ideation, methodological design, experimental execution, or result analysis. All scientific contributions and substantive writing were carried out by the authors.

### ACKNOWLEDGMENTS

This project is supported by Fundamental and Interdisciplinary Disciplines Breakthrough Plan of the Ministry of Education of China (No. JYB2025XDXM101), NSF of China Projects (Nos. 62550004, U25B6003, 92370124, 92248303); Beijing Natural Science Foundation L247011; the High Performance Computing Center, Tsinghua University. J.Z was also supported by the XPlorer Prize.

We especially thank Lifan Yuan for collaboration. We thank Wei Xiong, Zekun Hao, Yuxuan Tong, Chang Zhou, and Jiashu Xu for the insightful discussion.

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

# A    PROOF OF THEOREMS

**Theorem A.1** (**Policy Optimization with Negative Answers**). *Consider the maximum-likelihood objective for training the implicit negative policy* $\pi_\theta^-$:

$$\max_\theta \mathbb{E}_{p(q)\pi^-(a|q)}\left[\log \pi_\theta^-(a|q)\right] \Leftrightarrow \min_\theta \left[-\mathbb{E}_{(q,a)\sim\mathcal{D}^-}\log \frac{\pi_{old}(a|q) - r_q \pi_\theta^+(a|q)}{1 - r_q}\right]$$

*Assuming unlimited data and model capacity, the optimal solution for solving Eq. 8 is*

$$\forall q, a: \quad \pi_\theta^+(a|q) = \pi^+(a|q)$$

*Proof.* The proof is quite straightforward. First, we show that maximum-likelihood training leads to the optimal solution $\pi_{\theta^*}^-(a|q) = \pi^-(a|q)$.

$$\max_\theta \mathbb{E}_{p(q)\pi^-(a|q)}\left[\log \pi_\theta^-(a|q)\right]$$

$$\Leftrightarrow \max_\theta \mathbb{E}_{p(q)\pi^-(a|q)}\left[\log \pi_\theta^-(a|q) - \log \pi^-(a|q)\right]$$

$$\Leftrightarrow \min_\theta \mathbb{E}_{p(q)} D_{\mathrm{KL}}\left[\pi^-(a|q)\|\pi_\theta^-(a|q)\right]$$

Since $D_{\mathrm{KL}}\left[\pi^-(a|q)\|\pi_\theta^-(a|q)\right] \geq 0$. The equality holds if and only if $\forall q : \pi_\theta^- = \pi^-$. We thus have

$$\forall q, a: \quad \pi_{\theta^*}^-(a|q) = \pi^-(a|q). \tag{11}$$

Next, we prove $\pi_{\theta^*}^+ = \pi^+$.

Note that that during training, $\pi_\theta^-$ is only an *implicit* policy defined by $\pi_\theta^+$ through

$$\pi_\theta^-(a|q) := \frac{\pi_{\mathrm{old}}(a|q) - r_q \pi_\theta^+(a|q)}{1 - r_q}.$$

On the other hand, the negative data distribution $\pi^-$ has the same relationship with $\pi^+$ by Eq. 7.

$$\pi^-(a|q) := \frac{\pi_{\mathrm{old}}(a|q) - r_q \pi^+(a|q)}{1 - r_q}.$$

We have ensured $0 < r_q < 1$ during training, combining these observations and Eq. 11, we have

$$\forall q, a: \quad \pi_{\theta^*}^+(a|q) = \pi^+(a|q)$$

$\square$

**Proposition A.2** (**Algorithm Gradient Comparision**). *Suppose there are $\hat{r}_q K$ positive answers and $(1 - \hat{r}_q)K$ negative ones for a given question $q$*

*(a) GRPO Gradient: Consider only binary reward $\{0, 1\}$ in Eq. 3, GRPO loss gradient satisfies*

$$\nabla_\theta \mathcal{L}_\mathcal{D}^{GRPO}(\theta) = -\sum \left\{rA_q^+ \cdot \mathcal{I}\left[R_\theta^t(q,a) < 1+\epsilon'\right] + (1-r)A_q^- \cdot \mathcal{I}\left[R_\theta^t(q,a) > 1-\epsilon'\right]\right\}\nabla_\theta R_\theta^t(q,a),$$

*where $A_q^+ = \sqrt{\frac{1-\hat{r}_q}{\hat{r}_q}}$ and $A_q^- = -\sqrt{\frac{\hat{r}_q}{1-\hat{r}_q}}$ are respectively normalized advantages for answers.*

*(b) NFT Gradient: Let $\omega(q) = \sqrt{(1-\hat{r}_q)/\hat{r}_q}$, NFT loss gradient satisfies*

$$\nabla_\theta \mathcal{L}_\mathcal{D}^{NFT}(\theta) = -\sum \left\{rA_q^+ \cdot \frac{1}{R_\theta^t(q,a)} + (1-r)A_q^- \cdot \max\left[\frac{1-\hat{r}_q}{1-\hat{r}_q} \frac{R_\theta^t(q,a)}{}, \epsilon\right]^{-1}\right\}\nabla_\theta R_\theta^t(q,a).$$

*Proof.* **(a) GRPO Gradient:** We first copy down the GRPO loss definition from Eq. 3.

$$\mathcal{L}_\mathcal{D}^{\mathrm{GRPO}}(\theta) = -\sum_{q,a,t} \min\left[R_\theta^t(q,a)\hat{A}_{q,a}, \mathrm{clip}(R_\theta^t(q,a), 1-\epsilon', 1+\epsilon')\hat{A}_{q,a}\right].$$

The normalized advantage can be estimated as

$$\hat{A}_{\boldsymbol{q},\boldsymbol{a}} := \left[r(\boldsymbol{q},\boldsymbol{a}) - \texttt{mean}\{r^{1:K}\}\right] / \texttt{std}\{r^{1:K}\}$$

$$\text{where} \quad \texttt{mean}\{r^{1:K}\} = \frac{1}{K}\left[\hat{r}_{\boldsymbol{q}}K * 1 + (1-\hat{r}_{\boldsymbol{q}})K * 0\right] = \hat{r}_{\boldsymbol{q}}$$

$$\text{and} \quad \texttt{std}\{r^{1:K}\} = \sqrt{\frac{1}{K}\left[\hat{r}_{\boldsymbol{q}}K * (1-\hat{r}_{\boldsymbol{q}})^2 + (1-\hat{r}_{\boldsymbol{q}})K * (0-\hat{r}_{\boldsymbol{q}})^2\right]} = \sqrt{\hat{r}_{\boldsymbol{q}}(1-\hat{r}_{\boldsymbol{q}})}.$$

When $\boldsymbol{a}$ is a positive answer, $r(\boldsymbol{q},\boldsymbol{a}) = 1$. We have $A_{\boldsymbol{q}}^+ = \sqrt{\frac{1-\hat{r}_{\boldsymbol{q}}}{\hat{r}_{\boldsymbol{q}}}} > 0$.

$$\mathcal{L}_{\mathcal{D}^+}^{\text{GRPO}}(\theta) = -\sum_{\boldsymbol{q},\boldsymbol{a}^+,t} \min\left[R_\theta^t(\boldsymbol{q},\boldsymbol{a}^+), 1+\epsilon'\right]\hat{A}_{\boldsymbol{q},\boldsymbol{a}^+}$$

$$\nabla_\theta \mathcal{L}_{\mathcal{D}^+}^{\text{GRPO}}(\theta) = -\sum_{\boldsymbol{q},\boldsymbol{a}+,t} A_{\boldsymbol{q}}^+ \cdot \mathcal{I}\left[R_\theta^t(\boldsymbol{q},\boldsymbol{a}^+) < 1+\epsilon'\right]. \tag{12}$$

When $\boldsymbol{a}$ is a negative answer, $r(\boldsymbol{q},\boldsymbol{a}) = 0$. We have $A_{\boldsymbol{q}}^- = -\sqrt{\frac{\hat{r}_{\boldsymbol{q}}}{1-\hat{r}_{\boldsymbol{q}}}} < 0$.

$$\mathcal{L}_{\mathcal{D}^-}^{\text{GRPO}}(\theta) = -\sum_{\boldsymbol{q},\boldsymbol{a}^-,t} \max\left[R_\theta^t(\boldsymbol{q},\boldsymbol{a}^-), 1-\epsilon'\right]\hat{A}_{\boldsymbol{q},\boldsymbol{a}^-}$$

$$\nabla_\theta \mathcal{L}_{\mathcal{D}^-}^{\text{GRPO}}(\theta) = -\sum_{\boldsymbol{q},\boldsymbol{a}^-,t} A_{\boldsymbol{q}}^- \cdot \mathcal{I}\left[R_\theta^t(\boldsymbol{q},\boldsymbol{a}^-) > 1-\epsilon'\right]. \tag{13}$$

Combining Eq. 12 and Eq. 13, we have

$$\nabla_\theta \mathcal{L}_{\mathcal{D}}^{\text{GRPO}}(\theta) = -\sum \left\{ rA_{\boldsymbol{q}}^+ \cdot \mathcal{I}\left[R_\theta^t(\boldsymbol{q},\boldsymbol{a}) < 1+\epsilon'\right] + (1-r)A_{\boldsymbol{q}}^- \cdot \mathcal{I}\left[R_\theta^t(\boldsymbol{q},\boldsymbol{a}) > 1-\epsilon'\right] \right\} \nabla_\theta R_\theta^t(\boldsymbol{q},\boldsymbol{a}),$$

**(a) NFT Gradient:** We copy down the NFT loss definition from Eq. 10.

$$\mathcal{L}_{\mathcal{D}}^{\text{NFT}}(\theta) = -\sum_{\boldsymbol{q},\boldsymbol{a},t} \omega(\boldsymbol{q})\left[r\log R_\theta^t(\boldsymbol{q},\boldsymbol{a}) + (1-r)\log \max\_v(\frac{1-\hat{r}_{\boldsymbol{q}}}{1-\hat{r}_{\boldsymbol{q}}} R_\theta^t(\boldsymbol{q},\boldsymbol{a}), \epsilon)\right]$$

When $\boldsymbol{a}$ is a positive answer, $r(\boldsymbol{q},\boldsymbol{a}) = 1$.

$$\mathcal{L}_{\mathcal{D}^+}^{\text{NFT}}(\theta) = -\sum_{\boldsymbol{q},\boldsymbol{a}^+,t} \omega(\boldsymbol{q})\log R_\theta^t(\boldsymbol{q},\boldsymbol{a})$$

$$= -\sum_{\boldsymbol{q},\boldsymbol{a}^+,t} \sqrt{\frac{1-\hat{r}_{\boldsymbol{q}}}{\hat{r}_{\boldsymbol{q}}}} \log R_\theta^t(\boldsymbol{q},\boldsymbol{a})$$

$$= -\sum_{\boldsymbol{q},\boldsymbol{a}^+,t} A_{\boldsymbol{q}}^+ \log R_\theta^t(\boldsymbol{q},\boldsymbol{a}^+)$$

$$\nabla_\theta \mathcal{L}_{\mathcal{D}^+}^{\text{NFT}} = -\sum_{\boldsymbol{q},\boldsymbol{a}^+,t} A_{\boldsymbol{q}}^+ \frac{1}{R_\theta^t(\boldsymbol{q},\boldsymbol{a}^+)} \nabla_\theta R_\theta^t(\boldsymbol{q},\boldsymbol{a}^+) \tag{14}$$

When $\boldsymbol{a}$ is a negative answer, $r(\boldsymbol{q},\boldsymbol{a}) = 0$.

$$\mathcal{L}_{\mathcal{D}^-}^{\text{NFT}}(\theta) = -\sum_{\boldsymbol{q},\boldsymbol{a}^-,t} \omega(\boldsymbol{q})\log\left[\max\_v(\frac{1-\hat{r}_{\boldsymbol{q}}}{1-\hat{r}_{\boldsymbol{q}}} R_\theta^t(\boldsymbol{q},\boldsymbol{a}^-), \epsilon)\right]$$

$$= -\sum_{\boldsymbol{q},\boldsymbol{a}^-,t} \sqrt{\frac{1-\hat{r}_{\boldsymbol{q}}}{\hat{r}_{\boldsymbol{q}}}} \log\left[\max\_v(\frac{1-\hat{r}_{\boldsymbol{q}}}{1-\hat{r}_{\boldsymbol{q}}} R_\theta^t(\boldsymbol{q},\boldsymbol{a}^-), \epsilon)\right]$$

$$\nabla_\theta \mathcal{L}_{\mathcal{D}^-}^{\text{NFT}} = - \sum_{\boldsymbol{q}, \boldsymbol{a}^-, t} \sqrt{\frac{1 - \hat{r}_{\boldsymbol{q}}}{\hat{r}_{\boldsymbol{q}}}} \left[ \max\left(\frac{1 - \hat{r}_{\boldsymbol{q}} R_\theta^t(\boldsymbol{q}, \boldsymbol{a}^-)}{1 - \hat{r}_{\boldsymbol{q}}}, \epsilon\right)^{-1} \cdot \frac{-\hat{r}_{\boldsymbol{q}}}{1 - \hat{r}_{\boldsymbol{q}}} \cdot \nabla_\theta R_\theta^t(\boldsymbol{q}, \boldsymbol{a}^-) \right]$$

$$= - \sum_{\boldsymbol{q}, \boldsymbol{a}^-, t} -\sqrt{\frac{\hat{r}_{\boldsymbol{q}}}{1 - \hat{r}_{\boldsymbol{q}}}} \left[ \max\left(\frac{1 - \hat{r}_{\boldsymbol{q}} R_\theta^t(\boldsymbol{q}, \boldsymbol{a}^-)}{1 - \hat{r}_{\boldsymbol{q}}}, \epsilon\right)^{-1} \cdot \nabla_\theta R_\theta^t(\boldsymbol{q}, \boldsymbol{a}^-) \right]$$

$$= - \sum_{\boldsymbol{q}, \boldsymbol{a}^-, t} A_{\boldsymbol{q}}^- \left[ \max\left(\frac{1 - \hat{r}_{\boldsymbol{q}} R_\theta^t(\boldsymbol{q}, \boldsymbol{a}^-)}{1 - \hat{r}_{\boldsymbol{q}}}, \epsilon\right)^{-1} \cdot \nabla_\theta R_\theta^t(\boldsymbol{q}, \boldsymbol{a}^-) \right] \tag{15}$$

Combining Eq. 14 and Eq. 15, we have

$$\nabla_\theta \mathcal{L}_{\mathcal{D}}^{\text{NFT}}(\theta) = - \sum \left\{ r A_{\boldsymbol{q}}^+ \cdot \frac{1}{R_\theta^t(\boldsymbol{q}, \boldsymbol{a})} + (1 - r) A_{\boldsymbol{q}}^- \cdot \max\left[\frac{1 - \hat{r}_{\boldsymbol{q}} R_\theta^t(\boldsymbol{q}, \boldsymbol{a})}{1 - \hat{r}_{\boldsymbol{q}}}, \epsilon\right]^{-1} \right\} \nabla_\theta R_\theta^t(\boldsymbol{q}, \boldsymbol{a}).$$

$\square$

*Remark* A.3 (**Dr. GRPO**). The main practical difference between Dr. GRPO (Liu et al., 2025b) and GRPO is that Dr. GRPO removes the std normalization term when estimating group-normalized advantages. Following Proposition 4.1, we simply need to set $\omega(\boldsymbol{q}) = 1 - \hat{r}_{\boldsymbol{q}}$ instead of $\omega(\boldsymbol{q}) = \sqrt{\frac{1 - \hat{r}_{\boldsymbol{q}}}{\hat{r}_{\boldsymbol{q}}}}$ to align with the Dr. GRPO loss function.

**Proposition A.4 (On-policy Gradient Equivalence).** *Following the set up of Proposition 4.1 and let $\epsilon \leq 1$, GRPO and NFT loss gradient are equivalent in on-policy training:*

$$R_\theta^t(\boldsymbol{q}, \boldsymbol{a}) = 1 \implies \nabla_\theta \mathcal{L}_{\mathcal{D}}^{\text{NFT}}(\theta) = \nabla_\theta \mathcal{L}_{\mathcal{D}}^{\text{GRPO}}(\theta)$$

*Proof.* The proof is simple. When on-policy, $R_\theta^t(\boldsymbol{q}, \boldsymbol{a}) = 1$.

Regarding positive answers $a^+$, GRPO gradient (Eq. 12) and NFT gradient (Eq. 14) both become

$$\nabla_\theta \mathcal{L}_{\mathcal{D}^+}^{\text{GRPO}}(\theta) = \nabla_\theta \mathcal{L}_{\mathcal{D}^+}^{\text{NFT}}(\theta) = A_{\boldsymbol{q}}^+ \nabla_\theta R_\theta^t(\boldsymbol{q}, \boldsymbol{a}^+).$$

Regarding negative answers $a^-$, GRPO gradient (Eq. 13) and NFT gradient (Eq. 15) both become

$$\nabla_\theta \mathcal{L}_{\mathcal{D}^-}^{\text{GRPO}}(\theta) = \nabla_\theta \mathcal{L}_{\mathcal{D}^-}^{\text{NFT}}(\theta) = A_{\boldsymbol{q}}^- \nabla_\theta R_\theta^t(\boldsymbol{q}, \boldsymbol{a}^-).$$

$\square$

# B DISCUSSION FOR CONTINUOUS REWARDS

In this section, we explain how NFTs can be extended to continuous reward settings.

First, we define $r := p(\mathbf{o} = 1|\boldsymbol{q}, \boldsymbol{a}) \in [0, 1]$, where $\mathbf{o} = 1$ means the answer is an optimal answer.

Then we define positive and negative distributions.

$$\pi^+(\boldsymbol{a}|\boldsymbol{q}) := \pi(\boldsymbol{a}|\boldsymbol{q}, \mathbf{o}=1) = \frac{\pi_{\text{old}}(\boldsymbol{a}|\boldsymbol{q}) p(\mathbf{o}=1|\boldsymbol{q}, \boldsymbol{a})}{\sum_A \pi_{\text{old}}(\boldsymbol{a}|\boldsymbol{q}) p(\mathbf{o}=1|\boldsymbol{q}, \boldsymbol{a})} \propto \pi_{\text{old}}(\boldsymbol{a}|\boldsymbol{q}) r(\boldsymbol{q}, \boldsymbol{a}),$$

$$\pi^-(\boldsymbol{a}|\boldsymbol{q}) := \pi(\boldsymbol{a}|\boldsymbol{q}, \mathbf{o}=0) = \frac{\pi_{\text{old}}(\boldsymbol{a}|\boldsymbol{q}) p(\mathbf{o}=0|\boldsymbol{q}, \boldsymbol{a})}{\sum_A \pi_{\text{old}}(\boldsymbol{a}|\boldsymbol{q}) p(\mathbf{o}=0|\boldsymbol{q}, \boldsymbol{a})} \propto \pi_{\text{old}}(\boldsymbol{a}|\boldsymbol{q})[1 - r(\boldsymbol{q}, \boldsymbol{a})],$$

$$r_{\boldsymbol{q}} \pi^+(\boldsymbol{a}|\boldsymbol{q}) + [1 - r_{\boldsymbol{q}}] \pi^-(\boldsymbol{a}|\boldsymbol{q}) = \pi_{\text{old}}(\boldsymbol{a}|\boldsymbol{q}),$$

Finally we copy training objective Eq. 9 below.

$$\mathcal{L}_{(\boldsymbol{a}, \boldsymbol{q}, r) \sim \mathcal{D}}^{\text{NFT}}(\theta) = r \left[ -\log \frac{\pi_\theta^+(\boldsymbol{a}|\boldsymbol{q})}{\pi_{\text{old}}(\boldsymbol{a}|\boldsymbol{q})} \right] + (1 - r) \left[ -\log \frac{1 - r_{\boldsymbol{q}} \frac{\pi_\theta^+(\boldsymbol{a}|\boldsymbol{q})}{\pi_{\text{old}}(\boldsymbol{a}|\boldsymbol{q})}}{1 - r_{\boldsymbol{q}}} \right] \tag{16}$$

**Theorem B.1 (NFT for continuous rewards).** *Consider the training objective of Eq. 16 with arbitrary continuous rewards $r(\boldsymbol{q}, \boldsymbol{a}) \in [0, 1]$. Suppose we can accurately estimate $r_{\boldsymbol{q}} = \mathbb{E}_{\pi_{old}(\boldsymbol{a}|\boldsymbol{q})} r(\boldsymbol{q}, \boldsymbol{a})$. Assuming unlimited data and model capacity, the optimal solution satisfies*

$$\forall \boldsymbol{q}, \boldsymbol{a} : \quad \pi_{\theta^*}^+(\boldsymbol{a}|\boldsymbol{q}) = \pi^+(\boldsymbol{a}|\boldsymbol{q})$$

*Proof.*

$$\nabla_\theta \mathcal{L}^{\text{NFT}}_{(\boldsymbol{a},\boldsymbol{q},r)\sim\mathcal{D}}(\theta) = \nabla_\theta \left[ r(\boldsymbol{q},\boldsymbol{a}) \left[ -\log \pi_\theta^+(\boldsymbol{a}|\boldsymbol{q}) \right] + (1 - r(\boldsymbol{q},\boldsymbol{a})) \left[ -\log \frac{\pi_{\text{old}}(\boldsymbol{a}|\boldsymbol{q}) - r_{\boldsymbol{q}}\pi_\theta^+(\boldsymbol{a}|\boldsymbol{q})}{1 - r_{\boldsymbol{q}}} \right] \right]$$

$$= \left[ r(\boldsymbol{q},\boldsymbol{a}) \left[ -\frac{1}{\pi_\theta^+(\boldsymbol{a}|\boldsymbol{q})} \right] + (1 - r(\boldsymbol{q},\boldsymbol{a})) \left[ \frac{r_{\boldsymbol{q}}}{\pi_{\text{old}}(\boldsymbol{a}|\boldsymbol{q}) - r_{\boldsymbol{q}}\pi_\theta^+(\boldsymbol{a}|\boldsymbol{q})} \right] \right] \nabla_\theta \pi_\theta(\boldsymbol{a}|\boldsymbol{q})$$

$$= \left[ \frac{r_{\boldsymbol{q}}\pi_\theta^+(\boldsymbol{a}|\boldsymbol{q}) - r(\boldsymbol{q},\boldsymbol{a})\pi_{\text{old}}(\boldsymbol{a}|\boldsymbol{q})}{\pi_\theta^+(\boldsymbol{a}|\boldsymbol{q})[\pi_{\text{old}}(\boldsymbol{a}|\boldsymbol{q}) - r_{\boldsymbol{q}}\pi_\theta^+(\boldsymbol{a}|\boldsymbol{q})]} \right] \nabla_\theta \pi_\theta(\boldsymbol{a}|\boldsymbol{q})$$

When $r_{\boldsymbol{q}}\pi_\theta^+(\boldsymbol{a}|\boldsymbol{q}) - r(\boldsymbol{q},\boldsymbol{a})\pi_{\text{old}}(\boldsymbol{a}|\boldsymbol{q}) = 0$, that is

$$\pi_\theta^+(\boldsymbol{a}|\boldsymbol{q}) = \frac{r(\boldsymbol{q},\boldsymbol{a})\pi_{\text{old}}(\boldsymbol{a}|\boldsymbol{q})}{r_{\boldsymbol{q}}} = \pi^+(\boldsymbol{a}|\boldsymbol{q})$$

We have $\nabla_\theta \mathcal{L}^{\text{NFT}}_{(\boldsymbol{a},\boldsymbol{q},r)\sim\mathcal{D}}(\theta) = 0$ constantly holds. $\square$

## C  EXPERIMENT DETAILS

**General training setup.** All algorithms are implemented based on the official DAPO codebase within the VeRL framework. We use a learning rate of 1e-6 with a linear warm-up schedule across all experiments. At each rollout step, we generate 16 answers for each of 512 sampled questions, then split the data into 16 mini-batches and train the policy network for 16 gradient steps. Models are trained for 320 rollout steps, totaling over 5,000 gradient steps. Unless otherwise specified, we follow DAPO's default design choices, including dynamic data sampling, token-level loss normalization, and no KL regularization. For 7B training, we restrict context lengths to 4K and use 64 NVIDIA H100 GPUs. For 32B training, we restrict context lengths to 32K (DAPO) or 16K (others), and use 128-256 NVIDIA H100 GPUs.

**DAPO.** is a variant of GRPO that has achieved state-of-the-art AIME performance on 32B models. Our NFT implementation is adapted from the official DAPO codebase, based on the VeRL framework (Sheng et al., 2024). NFT inherits most of DAPO's hyperparameters and design choices, including dynamic data sampling, token-level loss normalization, and no KL regularization. We adopt a faithful implementation of the official DAPO codebase, keeping all hyperparameters untouched.

**NFT.** Compared with DAPO, NFT modifies the context length to 16K for 32B training. We find this does not significantly affect performance, but noticeably reduces data collection time. Another difference is that we remove the overlong reward shaping technique of DAPO to ensure a binary reward outcome. In our setting, truncated answers are treated as negative, which sufficiently discourages overlong answers. The negative ratio clipping parameter is set to $\epsilon = 1.0$, and the prompt weight is defined as $\omega(\boldsymbol{q}) = 1 - r_{\boldsymbol{q}}$.

**RFT** is a simple but effective SL baseline that only fine-tunes LLMs on positive answers and throws away negative data. In our implementation, the main difference between RFT and NFT is that RFT zeros out negative data loss and keeps a constant prompt weight $\omega(\boldsymbol{q}) = 1$ during training. For RFT, we zero out negative data loss in NFT implementation and keep a constant prompt weight $\omega(\boldsymbol{q}) = 1$ during training.

**GRPO** does not adopt the dynamic sampling technique proposed by DAPO. Rather, it simply leverages all data for training, even though the gradient for all-positive or all-negative questions should be zeroed out automatically by the algorithm itself (Yu et al., 2025). Other DAPO-related techniques are kept, such as setting positive clipping parameter to a higher value $\epsilon'_+ = 0.28$. Since GRPO requires less time for sampling data, we train GRPO models for 580+ rollout steps instead of 320 steps, which roughly takes the same training time compared with DAPO experiments.

**Dr. GRPO** is modified from our GRPO implementation. The only difference is the removal of the std normalization when computing group-normalized advantages.

**Iterative DPO.** Comparing DPO with other RL algorithms in a head-to-head fashion is difficult because DPO requires paired data to calculate the training objective, while we sample 16 unpaired

answers for each question. To solve this problem, we take the implementation of InfoNCA (Chen et al., 2024), a variation of the DPO algorithm that can handle $K > 2$ responses per question:

$$\mathcal{L}^{\text{InfoNCA}}_{(\boldsymbol{q},\boldsymbol{a}^{1:K},r^{1:K})\sim\mathcal{D}}(\theta) = -\sum_{k=1}^{K}\left[\frac{r^{(k)}}{\sum_{i=1}^{K}r^{(i)}}\log\frac{e^{\beta R_\theta(\boldsymbol{q},\boldsymbol{a}^k)}}{\sum_{i=1}^{K}e^{\beta R_\theta(\boldsymbol{q},\boldsymbol{a}^i)}}\right]$$

InfoNCA is guaranteed to become DPO algorithm in $K = 2$ settings. We ablate $\beta \in \{0.1, 0.01, 0.02\}$ and report the best averaged validation result. We find InfoNCA training to be unstable and add an SFT regularization term to the original loss function for stabilizing the training process.

**Validation details.** Validation is performed with a top-p value of $0.7$. The temperature is set to $1.0$ for 7B models and $0.6$ for 32B models, with context lengths matching the training configuration. We use math-verify (Kydlíček, 2024) as the verifier during training validation, and simpleRL (Zeng et al., 2025) for final evaluation. The DAPO-17k benchmark consists solely of training questions whose ground truth answers are integers and includes both a prefix and a lastfix for each question. Accordingly, for validation on AIME and AMC questions, we adapt the prompt format to match the training pattern. For other benchmarks with non-integer answers, question prompts remain unmodified.

## D  ADDITIONAL EXPERIMENT RESULTS

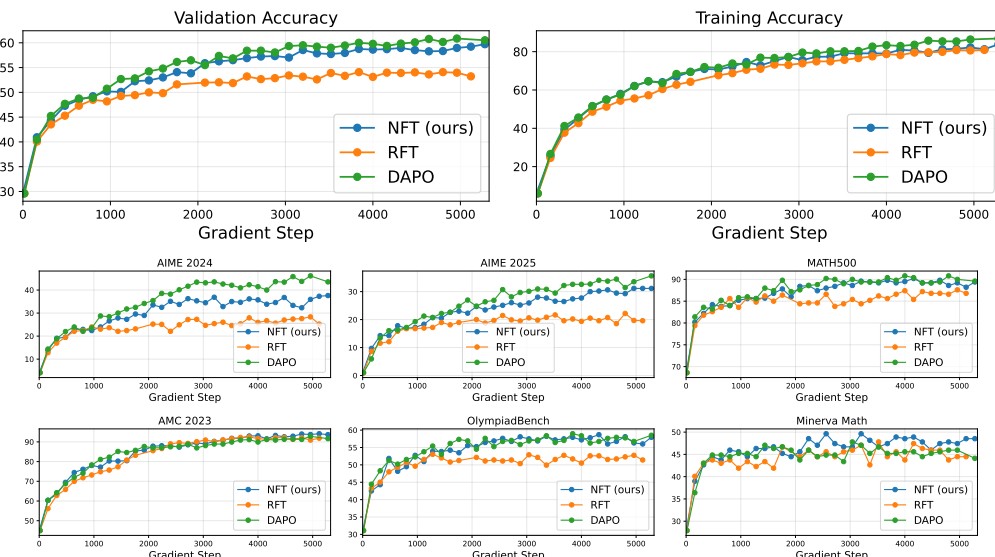

Figure 11: Training and validation accuracy curves for 32B experiments.

