# OpenReview forum: "NFT: Bridging Supervised Learning and Reinforcement Learning in Math Reasoning"
_ICLR.cc/2026/Conference — ICLR 2026 Poster_

### Official Review · Reviewer_Nkge · 2025-11-01

**Soundness:** 2
**Presentation:** 3
**Contribution:** 2
**Rating:** 4
**Confidence:** 3

**Summary:**

This paper proposes Negative-aware Fine-Tuning (NFT), a new supervised learning algorithm designed to bridge the conceptual and empirical gap between SL and RL in math reasoning.

Unlike traditional SL methods that discard incorrect generations, NFT explicitly models negative data by introducing an implicit negative policy parameterized by the same LLM. The paper shows that NFT can optimize both positive and negative samples via a single model, with negligible memory cost. Empirically, experiments on Qwen2.5-Math-7B and 32B show that NFT outperforms or matches leading RL baselines like GRPO and DAPO on math reasoning benchmarks (AIME24/25, AMC23, MATH500, OlympiadBench, Minerva).

**Strengths:**

1. The theoretical analysis is interesting to bridge the gap between GRPO and NFT.

2. The empirical observations are interesting. The analysis of entropy trends and scaling effects (negative feedback being more important for larger models) adds valuable interpretive depth.

3. The writing is good and easy to follow.

**Weaknesses:**

1. It is well known that incorporating negative examples can enhance model performance, both in offline and online settings. Therefore, the observed performance improvement from training with negative data is not particularly surprising.

2. I am not sure how the authors define SFT, as the proposed method seems more consistent with an RL training setup. I am not fully convinced that NFT should be categorized as a purely supervised fine-tuning (SFT) algorithm. It would be great if the authors can further elaborate on this.

3. From Figure 1, it appears that the main difference between NFT and GRPO lies in the use of binary labels, which essentially changes how the advantage of responses is computed.

4. It seems that NFT does not outperform DAPO (even slightly worse). What is the key advantage of NFT compared to GRPO-like algorithms such as DAPO?  What can drive people to use NFT rather than DAPO?

**Questions:**

Please see the weakness.

---

> ### Author Response · Authors · 2025-11-18
> **Official Response to Reviewer Nkge (part 1/2)**
>
> We thank the reviewer for their time. We hope our response below can address the reviewer's concerns.
>
>
> **Q1: The observed performance improvement from negative data is not particularly surprising.**
>
> **A1**: Different from all previous works. **The core contribution of NFT lies in proving that we can achieve training convergence on negative data.**
>
> For easier understanding, let's compare it with Unlikelihood Training.
>
> $\mathcal{L}^\text{unlikelihood}(\theta) := -\mathbb{E}\_{x \sim \mathcal{D^+}}  \left[ \log\pi\_\theta(y \mid x)\right]  +\mathbb{E}\_{x \sim \mathcal{D^-}}  \left[ \log\pi\_\theta(y \mid x)\right]$
>
> $\mathcal{L}^\text{NFT}(\theta) := -\mathbb{E}\_{x \sim \mathcal{D^+}}  \left[ \log\pi\_\theta(y \mid x) \right]  -\mathbb{E}\_{x \sim \mathcal{D^-}}  \left[ \log\pi\_\theta^-(y \mid x) \right]$
>
> where  $\pi^-\_\theta$ is an **implicit** policy: $\pi^-\_\theta := \frac{\pi^\text{old}(x,y) - r\_q \pi\_\theta^+(x,y)}{1-r\_q}$.
>
> These two objectives handle negative data in fundamentally different ways.
>
> Consider a special case where we discard all positive data in $D^+$ and train the LLM **only** on the negative dataset $D^-$.
>
> **Unlikelihood:** Simply minimizes log likelihood. The loss will diverge, and there is no convergence guarantee.
>
> **NFT (ours):** It still maximizes the likelihood of $\pi^-\_\theta$, so we can guarantee convergence $\pi^-\_{\theta} = \pi^-$. Since $\pi^-\_\theta$ is an implicit policy, we are actually training the underlying $\pi\_\theta$. Theorem 3.1 in the paper proves convergence to the target negative distribution, i.e., $\pi_{\theta^*} = \pi^-$.
>
> In summary, NFT guarantees convergence even when trained only on negative data. We believe this finding is novel and contradicts the assumptions in much of the prior work. To prove this point, we present the following experimental results:
>
> | | 7B-Base | + RFT | +Unlikelihood | +NFT (neg. data only) | + NFT |
> | :--- | :---: | :---: | :---: | :---: | :---: |
> | positive data | no | yes | no | no | yes |
> | negative data | no | no | yes | yes | yes |
> | Avg. Math | 31.6 | 48.3 | Collapse | 44.2 | **51.7** |
>
>
>
> **Q2: How to define SFT, as the proposed method seems more consistent with an RL training setup. I am not fully convinced that NFT should be categorized as a SFT algorithm.**
>
> **A2**: NFT requires online data sampling and a reward model for data selection. If we focus on the **practical implementation**, we would agree with the reviewer that NFT can be categorized as an RL algorithm.
>
> However, we want to clarify that our classification is based on the algorithm's **theoretical derivation** rather than its practical implementation:
>
> In our work, "RL" refers to policy gradient methods, a line of research derived from the initial REINFORCE algorithm.
>
> In comparison, SL (rather than SFT) refers to the **maximum likelihood training** method.
>
> We believe both viewpoints are acceptable, and hope the reveiwer finds it acceptable we chose the latter notation to highlight that NFT and GRPO belong to different theoretical frameworks, **though the boundary between them is becoming increasingly blurred.** For instance, the online RFT algorithm presents a similar ambiguity: it can be viewed as supervised training based on its loss objective, or as reinforcement learning based on its training setup.
>
> We have updated Sec 2.1 of the paper to clarify the definitions of technical terms to avoid similar confusion. We sincerely thank the reviewer for these comments.

---

> ### Author Response · Authors · 2025-11-18
> **Official Response to Reviewer Nkge (part 2/2)**
>
> **Q3: What is the key advantage of NFT compared to GRPO-like algorithms such as DAPO?**
>
> **A3**: Our recent findings show that NFT has two key practical advantages: greater robustness to off-policy training and better generalizability.
>
> 1. **Off-policyness:** NFT and GRPO indeed perform similarly in on-policy training. However, when the training becomes more off-policy, NFT begins to show a clear advantage, which we attribute to the clear off-policy convergence guarantee provided by its supervised learning foundation.
>
> Off-policyness Definition:
> **Original**: At each training epoch: Collect K data points, and train for M gradient steps.
>
> **$\times n$ Off-policy**: At each training epoch: Collect $n*K$ data points, and train for $n*M$ gradient steps.
>
> A smaller $n$ is computationally more expensive because it requires more frequent switching between training and inference. However, it is well-known that a larger $n$ leads to a performance drop for algorithms like DAPO.
>
>
> | Off-policyness | $\times 1$ | $\times 2$ | $\times 4$ | $\times 16$ |
> | :--- | :---: | :---: | :---: | :---: |
> | DAPO Math Avg  | 51.2 | 50.1 | 37.2 | Collapsed |
> | NFT Math Avg   | 51.7 | 50.9 | 51.4 | 50.1 |
> | Efficiency     | +0.0% | +11.3% | +14.7% | +15.0% |
>
>
> 2. **Generalizability:** RFT/NFT benefit from an extremely simple formulation that **can be applied to almost any generative model beyond LLMs**. For instance, in image generation tasks, GRPO **cannot** be directly applied due to intractable likelihood issues. However, RFT can be directly applied to diffusion models using existing SL training methods.
>
> Although it is beyond the scope of this paper, we wish to mention **DiffusionNFT**, a follow-up work that extends NFT's principles to the diffusion domain. **DiffusionNFT achieves ~25 times greater efficiency compared with DiffusionGRPO.** We hope this successful application helps demonstrate the significant potential of our work, especially in unified RL across various modalities. This work marks the first step.
>
>
> **Q4: The main difference between NFT and GRPO lies in the use of binary labels, which essentially changes how the advantage of responses is computed?**
>
> **A4**: We respectfully disagree with this understanding.
>
> 1. The choice between binary or continuous reward is not the core difference between NFT and GRPO. In Appendix B of the paper, we prove that NFT can also be applied to continuous rewards. In short, this extends the framework beyond absolute correct (r=1) and incorrect (r=0) answers. For example, **a reward of r=0.8 means an answer has an 80% chance of being "correct" and a 20% chance of being "incorrect."**
> 2. The core difference, we believe, still lies in the theoretical properties of the two algorithms:
>
> - GRPO is derived from the REINFORCE algorithm, which is an on-policy method. It achieves off-policy training by applying importance sampling (IS).
> - NFT comes from the RFT algorithm; it can achieve off-policy training **without** leveraging IS. This supervised nature gives it more robustness and better results in off-policy settings, as shown in **A3**.

---

### Official Review · Reviewer_RLnb · 2025-11-01

**Soundness:** 3
**Presentation:** 4
**Contribution:** 4
**Rating:** 8
**Confidence:** 3

**Summary:**

The paper introduces Negative-aware Fine-Tuning (NFT), a supervised learning framework that allows large language models (LLMs) to learn from both correct and incorrect generations without RL. Unlike traditional supervised fine-tuning (or rejection finetuning), NFT constructs an implicit negative policy parameterized by the same model as the positive policy, enabling direct policy optimization on both positive and negative examples. The authors show theoretically that NFT is equivalent to RL under strict on-policy conditions, thereby bridging the conceptual and empirical gap between supervised and RLVR. Experiments on math reasoning tasks with Qwen-2.5 models (7B and 32B) demonstrate that NFT matches or surpasses leading RL methods like GRPO and DAPO while being simpler and more memory-efficient.

**Strengths:**

1. The paper is well written and clearly structured, making it easy to follow.

2. Theoretical contribution — the equivalence between NFT and GRPO is well derived and provides valuable conceptual clarity between reinforcement learning and supervised fine-tuning.

3. The method is simple, effective, and memory-efficient, avoiding the complexity of traditional RL pipelines.

4. Unlike DPO, NFT does not require explicit positive–negative pairs, which improves data efficiency and implementation flexibility.

5.  Empirical results demonstrate strong performance and good test-time scalability, with well-conducted analysis and ablation studies that support the claims.

**Weaknesses:**

1. The evaluation is limited to mathematical reasoning, with no experiments on logical or commonsense reasoning tasks, limiting evidence of generalization.

2. While Table 2 shows strong performance compared to other RL-based methods, the paper lacks quantitative analysis of efficiency (e.g., training FLOPs, wall-clock time, or GPU hours). Since NFT claims to be simpler and more efficient than RL-based training, this measurement is important to support the claim.

3. Additional discussion on failure cases would improve interpretability.

**Questions:**

1. Can NFT generalize beyond mathematical reasoning to less verifiable domains such as commonsense or code generation tasks?

2. Could the authors provide quantitative comparisons of training efficiency (e.g., FLOPs, GPU hours, or convergence speed) against RL-based baselines such as DAPO or GRPO?

3. Could the authors provide generation examples or failure cases?

---

> ### Author Response · Authors · 2025-11-18
> **Official Response to Reviewer RLnb**
>
> We thank the reviewer for the comments. We are glad the reviewer appreciates our work and have conducted experiments to address the reviewer's concerns below.
>
> **Q1: Experiments beyond mathematical reasoning (e.g., code /science).**
>
> **A1**: Following the reviewer's suggestion, we have conducted additional evaluations on coding and science benchmarks.
>
> (Code: LiveCodeBench, Science: GPQA and JEEBench)
> | | Math Avg | Science Avg | Code Avg |
> | :--- | :---: | :---: | :---: |
> | Base-model | 31.6 | 25.6 | 5.7 |
> | NFT        | **51.7** | **37.9** | **12.2** |
>
> The results show that although originally trained on math tasks, NFT generalizes well to other domains such as science and coding.
>
> For non-verifiable tasks such as common sense reasoning, we believe NFT can also be applied by employing existing reward models. Still, training with non-verifiable rewards remains an open challenge for the entire field.
>
> **Q2: Quantitative analysis of efficiency.**
>
> **A2**: We thank the reviewer for the helpful suggestion. We have measured the training efficiency of NFT and compared it with GRPO (which needs a reference model) and DAPO (which does not).
>
> | | GPU | train steps | wall time | memory/card | estimated FLOPs | performance |
> | :--- | :---: | :---: | :---: | :---: | :---: | :---: |
> | NFT  | 32 H100 | 5120 | 43h | 59.2 GB | 1.2e21 | 51.7 |
> | GRPO | 32 H100 | 5120 | 46h | 66.2 GB | 1.42e21 | 49.5 |
> | DAPO | 32 H100 | 5120 | 44h | 59.2 GB | 1.2e21 | 51.2 |
>
> In summary, NFT has better memory efficiency compared with GRPO by eliminating the need for a reference model as a regularizer.
>
>   **Q3: Generation examples or failure cases.**
>
> **A3**: Like many other RL algorithms, we observe that after tuning, NFT may sometimes generate overlong responses that exhibit a repetitive, never-ending pattern. Also, in some responses, the model might generate an excessive number of filler tokens like "wait" or "oh, I see". This is a generally known problem with RL algorithms, which can be largely mitigated by data filtering and distillation. Due to space limits, we have uploaded some failure cases into the supplementary material. We will incorporate a special case analysis into the Appendix should the reviewer find it necessary.

---

### Official Review · Reviewer_YUof · 2025-11-01

**Soundness:** 2
**Presentation:** 2
**Contribution:** 2
**Rating:** 2
**Confidence:** 4

**Summary:**

The paper introduces Negative-aware Fine-Tuning (NFT), a ``supervised'' method for large language model training with binary verifier feedback. Unlike rejection-based fine-tuning (RFT), which discards incorrect responses, NFT constructs an implicit negative policy that allows the model to learn from both correct and incorrect generations using the same parameterized model. The method defines a token-level objective combining positive and negative log-likelihood terms and incorporates clipping and straight-through gradients for stability. Conceptually, NFT connects supervised and reinforcement learning: under on-policy conditions, its gradients align with those of GRPO, while maintaining a maximum-likelihood foundation. Experiments on 7B and 32B models for math reasoning tasks show that NFT surpasses supervised baselines and achieves performance comparable to leading RL algorithms.

**Strengths:**

- Relevant and timely topic.
- Comprehensive evaluation across a broad range of reasoning datasets.

**Weaknesses:**

- The selling point of NFT is that it is an online SFT method. However, being ``online SFT'' does not immediately provide conceptual advantages. If one considers RL as a paradigm of learning under interaction with an environment, then online SFT is simply a special case of RL. The advantages therefore need to be shown practically, for example, through computational or empirical improvements, rather than assumed conceptually. The paper state:
    > Memory Efficiency. NFT is memory-efficient. In practice, we keep only a single model copy in memory. The old policy likelihood πold(a|q) can be pre-computed during data generation.

    However, this should also applies to GRPO (if KL-free) and DAPO.

- Empirically, NFT does not show notable differences from DAPO, suggesting that the distinction is modest in practice.

**Questions:**

- What confuses me is the following: what would happen if we simply optimize ($\beta = \pi_\text{old}$ due to rendering issue)

    $$\mathcal{L}(\theta) = \mathbb{E}_{x \sim \mathcal{D}, y \sim \beta} \left[ r(x, y) \left[ -\log\frac{\pi^\theta(y \mid x)}{\beta(y\mid x)} \right]  + (1- r(x, y) ) \left[ \log\frac{\pi^\theta(y \mid x)}{\beta(y\mid x)} \right] \right] $$

    This contrastive objective simply reduces the likelihood of negative samples while increasing that of positive ones.

    It would be helpful if the paper could clarify why Eq. (9) provides a more principled alternative to this naive approach, and in particular why the re-parameterization introduced in Eq. (7) is essential.

    An empirical comparison against this straightforward baseline, including the same practical tricks described in Section 3.3, would also make the claimed advantages of NFT more convincing.


#### minor comments: writing can be made more crisp


- Eq. (9): The notation $(a, q, r)\sim \mathcal{D}$ reads as if the dataset is static. It would help to clarify that $\mathcal{D}$ actually consists of online samples generated by $\pi_\text{old}$, rather than a fixed dataset.

- Algorithm 1: My intuition is that the training process should involve \emph{updating} $\pi_{\text{old}}$, yet this does not appear to be reflected in the pseudocode. If my intuition is correct, clarifying whether (and how) $\pi_{\text{old}}$ is updated across iterations would improve readability.

- The phrase ``a pretrained LLM $\pi_{\text{old}}$'' suggests that $\pi_{\text{old}}$ remains \emph{frozen}, tied to the initial pretrained model. It would be clearer to specify whether $\pi_{\text{old}}$ is only the initialization or if it is also updated online during training.

---

> ### Author Response · Authors · 2025-11-18
> **Official Response to Reviewer YUof (Part  1/2)**
>
> We thank the reviewer for the comments. We conduct new experiments to clarify our research motivation.
>
> **Q1: Why NFT provides a more principled alternative to a naive approach.**
>
> $\mathcal{L}^\text{naive}(\theta) = \mathbb{E}_{x \sim \mathcal{D}, y \sim \beta} \left[ r\left[ -\log\frac{\pi^\theta(y \mid x)}{\pi^\text{old}(y\mid x)} \right]  + (1- r) \left[ \log\frac{\pi^\theta(y \mid x)}{\pi^\text{old}(y\mid x)} \right] \right]$
>
> **A1**: In short, our method (NFT) can provide a clear convergence guarantee in off-policy settings, while the reviewer's proposal cannot.
>
> In $\mathcal{L}^\text{naive}(\theta)$ , **$\pi^\text{old}$ is redundant and can be removed.**
>
> $\mathcal{L}^\text{naive}(\theta)$ is essentially SFT + Unlikelihood training (i.e., minimizing likelihood for negative data).
> $\mathcal{L}^\text{naive}(\theta) \Leftrightarrow \mathbb{E}_{x \sim \mathcal{D}, y \sim \beta} \left[ r\left[ -\log\pi^\theta(y \mid x)\right]  + (1- r) \left[ \log\pi^\theta(y \mid x) \right] \right]$
>
> Hence, **there is no convergence guarantee for $\mathcal{L}^\text{naive}$.** The loss could diverge to infinity when training is off-policy.
>
> However, NFT **always maximizes likelihood** (using a standard CE loss) on positive and negative data, respectively.
> $\mathcal{L}^\text{NFT}(\theta) \Leftrightarrow \mathbb{E}_{x \sim \mathcal{D}, y \sim \beta} \left[ r\left[ -\log\pi\_\theta^+(y \mid x)\right]  + (1- r) \left[ - \log\pi^-\_\theta(y \mid x) \right] \right]$,
>
> where $\pi^-\_\theta$ is an **implicit** policy: $\pi^-\_\theta := \frac{\pi^\text{old}(x,y) - r_q \pi\_\theta^+(x,y)}{1-r_q}$.
>
> Since we only **maximize** likelihood, NFT can ensure convergence to the target distributions ($\pi\_{\theta}^- = \pi^-$ and thus $\pi\_{\theta^*}^+ = \pi^+$), even in off-policy settings. In contrast, $\mathcal{L}^\text{naive}$ cannot.
>
> **Q2: An empirical comparison against this straightforward baseline.**
>
> **A2**: We retained all the practical modifications from NFT-7B and only changed the loss objective:
>
> | Off-policyness | $\times 1$ | $\times 4$ |
> | :--- | :---: | :---: |
> | Naive Math Avg | 49.5 | Collapse |
> | NFT Math Avg   | **51.7** | **51.4** |
> | DAPO Math Avg  | 51.2 | 37.2 |
>
> NFT demonstrates a clear advantage.
>
> **Q3: The advantages therefore need to be shown practically ...... NFT does not show notable differences from DAPO.**
>
> **A3:** In our recent experiments, **NFT is more robust to off-policy training, clearly outperforming DAPO.**
>
> Definition:
> **Original**: At each training epoch: Collect K data points, and train for M gradient steps.
>
> **$\times n$ Off-policy**: At each training epoch: Collect $n*K$ data points, and train for $n*M$ gradient steps.
>
> A smaller $n$ is computationally more expensive because it requires more frequent switching between training and inference. However, it is well-known that a larger $n$ leads to a performance drop for algorithms like DAPO.
>
> | Off-policyness | $\times 1$ | $\times 2$ | $\times 4$ | $\times 16$ |
> | :--- | :---: | :---: | :---: | :---: |
> | DAPO Math Avg  | 51.2 | 50.1 | 37.2 | Collapsed |
> | NFT Math Avg   | 51.7 | 50.9 | 51.4 | 50.1 |
> | Efficiency     | +0.0% | +11.3% | +14.7% | +15.0% |
>
> Although NFT and DAPO perform similarly in near on-policy settings. NFT meets less severe performance decline when training is more off-policy. We attribute this gain to NFT's clear convergence guarantee.
>
> **Q4: Being online SFT does not immediately provide conceptual advantages. ....**
>
> **A4:** We believe online SFT (e.g., NFT) offers two conceptual advantages over Policy Gradient algorithms (e.g., DAPO). These initial observations motivated our study of NFT, even if it performs similarly to DAPO in some settings.
>
> 1. NFT achieves off-policy training without leveraging **importance sampling (IS)**. This stands in stark contrast to PPO, which patches on-policy RL algorithms like REINFORCE with token-level **IS**, adding variance to the gradient estimation. We believe this leads to better relative performance when training is more off-policy, as in **A2**
>
> 2. **generability:**  RFT/NFT enjoys extreme simplicity and better generability to **other generative models** beyond LLMs.   For instance, in image generation tasks, GRPO **cannot** be directly applied because **a diffusion model cannot calculate the likelihood $\log p(x)$ of an image.**[1] However, RFT can be directly applied to diffusion models using existing supervised training methods.
>
> Although it is beyond the scope of this paper, we wish to mention **DiffusionNFT**, a follow-up work that extends NFT's principles to the diffusion domain. **DiffusionNFT achieves ~25 times greater efficiency compared with DiffusionGRPO.** We hope this successful application helps demonstrate the significant potential of our work, especially in unified RL across various modalities. This work marks the first step.
>
> [1] DiffusionNFT: Online Diffusion Reinforcement with Forward Process

---

> ### Author Response · Authors · 2025-11-18
> **Official Response to Reviewer YUof (Part 2/2)**
>
> **Q5: If one considers RL as a paradigm of learning under interaction with an environment, online SFT is simply a special case of RL.**
>
> **A5**: We agree with the reviewer. NFT can be categorized as RL according to the reviewer's definition. NFT requires online data sampling and a reward model for data selection. If we focus on the **practical implementation**, it is indeed RL.
>
> However, we want to clarify that our classification is based on the algorithm's **theoretical derivation** rather than its practical implementation:
>
> In our work, "RL" refers to policy gradient methods, a line of research derived from the initial REINFORCE algorithm.
>
> In comparison, SL (rather than SFT) refers to the **maximum likelihood training** method.
>
> We believe both viewpoints are acceptable, and hope the reveiwer finds it acceptable we chose the latter notation to highlight that NFT and GRPO belong to different theoretical frameworks, **though the boundary between them is becoming increasingly blurred.** For instance, the online RFT algorithm presents a similar ambiguity: it can be viewed as supervised training based on its loss objective, or as reinforcement learning based on its training setup.
>
> We have updated Sec 2.1 of the paper to clarify the definitions of technical terms to avoid similar confusion. We sincerely thank the reviewer for these comments.
>
>
> **Q6: How $\pi^\text{old}$ is updated?  Is it frozen? NFT Online update?.**
>
> **A6**: We confirm that NFT is an **online** algorithm. $\pi^\text{old}$ is not frozen; it is updated at a fixed interval (after every 16 gradient steps), following the exact setup used in DAPO.
>
> In fact, NFT is more advantageous than DAPO in more off-policy scenarios (e.g., 64+ gradient update per epoch), as discussed in **A3**. This brings higher system efficiency, but we still align with DAPO's hyperparameters to maintain a fair comparison.
>
> We thank the reviewer for the detailed suggestion and will update the pseudocode to enhance clarity.

---

### Official Review · Reviewer_wqPT · 2025-11-03

**Soundness:** 2
**Presentation:** 3
**Contribution:** 2
**Rating:** 4
**Confidence:** 4

**Summary:**

This paper explores whether supervised fine-tuning approaches can be competetive with reinforcement learning. In particular, the authors identify a means of using negative samples to fine-tune an "implicit negative policy". Their approach successfully leverages both positive and negative samples, which can potentially improve over conventional supervised fine-tuning approaches that just use positive samples. Their results show some improvements over baselines, most notably in AMC23.

**Strengths:**

- The authors posit a clear research question to explore in their paper: the current approach to fine-tuning may be competitive for self-improvement compared to current approaches to reinforcement learning.
- Motivated by this research question, the authors identify a clear weakness in current apporaches to fine-tuning: it is not possible to use the full dataset when fine-tuning only on positive samples.

**Weaknesses:**

- While many aspects of the paper are clear, some of the benfits of their negative-aware fine tuning method are not obvious. For example, they choose to use a specific implicit policy approach, but dont compare against an explicit method that conditions on the reward. Also, in their ablations, their practical modifications seem to have a much larger effect on performance than solely using the negative data.
- The motivating research question does not feature prominently in the experiments, besides comparisons to reinforcement learning. That is, while supervised finetuning approaches appear competetive with reinforcement learning for performance improvement, this paper does not clearly demonstrate that this gap is closed by self-improvement from mistakes in negative data.

**Questions:**

- line 067: throughout the paper, you refer to an "implicit" negative policy. It would help to clarify early what it means for the policy to be implicit, and how this actually affects training the LLM
- Equation 7: Is the key observation being made in equation 7 is essentially the marginalization of $r$?
- Section 3.2: What is the advantage of this implicit formulation relative to training on an LLM that explicitly conditions on the reward? That is, training the positive policy as pi(a|q, r=1) and the negativev policy as pi(a|q, r=0).
- Theorem 3.1: This statement is essentially stating that the critical points are the same, which makes sense. But it is not quite clear what the advantage of this this particular formulation.
- Proposition 4.1: The similarity seems to be driven by the free parameter governing the prompt weighting, $\omega(q)$. Moreover, the differences pointed out in the gradients seem to originate in the practical modifications, rather than the core algorithm. As such, I am not sure whether these takeaways are related to negative-aware fine tuning in particular.
- Table 1: Results are a little inconclusive, where in some cases NFT offers a notable improvement (AMC23), but in others losing to RFT or RL based approaches
- Section 5.3 and Figure 8: I like the fact that the authors investigate the benefits of negative data, but the analysis here seems a little adhoc. For example, couldn't higher entropy be achieved by entropy regularization, which would not necessarily be helpful given that it doesnt use any negative data? The higher entropy could be explained by there being more data fit, which is harder than just fitting positive samples.
- Section 3.3 (practical modifications) and Section 5.4: I also appreciate that the authors investigate the importance of the practical modifications to the base algorithm. It is puzzling that prompt weighting improves accuracy for NFT well beyond RFT Would this seem to suggest that the modifications are more impactful than the negative data?

### Minor Comments
- Some ancillary comments do not seem to make any connection to the contribution, such as: "Later studies (Liu et al., 2025b) suggest removing the std term from Eq. 4"
- Notation is sometimes abused and informal, such as line 150: $\mathcal{D} \sim \pi$
- It is never explicitly defined that "positive answers" are correct answers.
- line 098: minor point but the larger issue in taking a gradient of the RL objective is actually that the parameter $\theta$ is involved in the sampling distribution rather than just in the log probability (as in supervised learning)
- Section 3: the paper would benefit from being more clear about what policy is being under consideration. For example, you refer to $\pi$ and $\pi_old$

---

> ### Author Response · Authors · 2025-11-18
> **# Official Response to Reviewer wqPT (part 1/2)**
>
> # Official Response to Reviewer wqPT
> We thank the reviewer for the detailed review. We hope we address the reviewer's concerns below:
>
> **Q1: It would help to clarify early what it means for the policy to be implicit, and how this actually affects training the LLM ....**
>
> **A1:** To achieve policy improvement, the previous method (RFT) only trains on correct (positive) answers and discards incorrect (negative) data.
>
> To leverage negative data, we also perform supervised tuning on them.
>
> $\max\_\theta  E_{\pi^\text{old}(y|x)}\log\pi^-\_\theta(y \mid x)$,
>
> However, $\pi^-\_\theta$ does not actually exist as a single explicit language model. Instead, $\pi^-\_\theta$ is defined by two existing LLMs, $\pi^\text{old}$ and $\pi^+_\theta$:
>
>  $\pi^-\_\theta$ is an **implicit** policy: $\pi^-\_\theta := \frac{\pi^\text{old}(x,y) - r_q \pi\_\theta^+(x,y)}{1-r_q}$
>
> As a result, we are actually optimizing $\pi\_\theta^+(x,y)$ while finetuning on negative data. The term **implicit negative policy** is inspired by the DPO algorithm, which employs an **implicit reward model**. The loss functions as a reward model training loss, but no explicit reward model exists; DPO essentially optimizes the underlying LLM policy.
>
> In Theorem 3.1 of the paper, we prove that this method ensures algorithmic improvement. We refer the reviewer to Lines 190-210 for details.
>
>
> **Q2: They choose to use a specific implicit policy approach, but don't compare against an explicit method that conditions on the reward. ....**
>
> **A2:** We present a comparison with one of the simplest forms below. This involves prompting the LLM to be aware of the answer's reward and then performing inference using only high-reward prompts.
>
>
> | | base | RFT | Reward Conditioned | NFT (ours) |
> | :--- | :---: | :---: | :---: | :---: |
> | Negative data? | None | No | Yes | Yes |
> | 7B   | 48.3 | 48.3 | 48.2 | 51.7 |
> | 32B  | 52.8 | 52.8 | 54.2 | 59.2 |
>
> **NFT shows significant advantage compared with explicit reward modeling methods**
>
> [1] Reward-Augmented Data Enhances Direct Preference Alignment of LLMs
> [2] Autonomous Learning From Success and Failure: Goal-Conditioned Supervised Learning with Negative Feedback
> [3] Reward-Conditioned Policies
>
> **Q3: What is the advantage of this implicit formulation relative to training on an LLM that explicitly conditions on the reward?**
>
> **A3:**
>
> 1. Our method does not require altering the prompt with a reward **condition**.
> 2. NFT directly optimizes the target policy (for r=1) using negative data (r=0). This contrasts with methods that would optimize a separate policy for r=0, which is then unused during evaluation.
> 3. As discussed and demonstrated in **A3**, NFT shows superior empirical results.
>
> **Q4: Practical modifications seem to have a much larger effect on performance than solely using the negative data.**
>
> **A4:** We respectfully disagree with the reviewer. We conducted more experiments to show that NFT is advantageous regardless of the practical modifications used.
>
> 1. Both NFT and RFT leverage the same practical modifications. They all benefit from them; however, it is clear that NFT achieves a higher performance ceiling (especially in our new 32B experiments).
>
> | | RFT-raw | RFT-tuned | NFT-raw | NFT-tuned |
> | :--- | :---: | :---: | :---: | :---: |
> | 7B   | 47.3 | 48.3 | 47.5 | **51.7** |
> | 32B  | 50.3 | 52.8 | 53.7 | **59.2** |
>
>
> 2. DAPO, GRPO, and other baselines we compare with also need practical tricks to be effective. GRPO requires clipping likelihood ratios, and DAPO needs dynamic data filtering. Without these modifications, all methods suffer from performance degradation.
>
> | | DAPO-raw | DAPO-tuned | NFT-raw | NFT-tuned |
> | :--- | :---: | :---: | :---: | :---: |
> | 7B   | 47.4 | 51.3 | 47.5 | **51.7** |
>
>
> In summary, the advantage of leveraging negative data is clear and significant. NFT outperforms RFT by a clear margin, achieving performance comparable to established RL algorithms. We have tried our best to make a fair comparison.

---

> ### Author Response · Authors · 2025-11-18
> **Official Response to Reviewer wqPT Part(2/2)**
>
> **Q5: Proposition 4.1 The similarity seems to be driven by the free parameter governing the prompt weighting. Differences pointed out in the gradients seem to originate in the practical modifications, rather than the core algorithm.**
>
> **A5**: We are sure this is **wrong understanding**. Proposition 4.1 is NOT governed by prompt weighting.
>
> The simplified loss gradient for NFT is
>  $\nabla\_\theta L^\text{NFT}\_\theta = w(q) [r A^+  * f^+(q,a) + (1-r) A^-* f^-(q,a)]$,
>  where $f^+$ and $f^-$ are some coefficient functions.
>
> RFT(SL) loss gradient:
>  $\nabla\_\theta L^\text{RFT}\_\theta =  w(q) [r A^+  * k^+(q,a)]$,
>
> GRPO loss gradient:
>  $\nabla\_\theta L^\text{GRPO}\_\theta =  [r A^+  * g^+(q,a) + (1-r) A^-* g^-(q,a)]$,
>
>
> As shown, the critical component of the NFT loss is the additional term related to $A^-$, which originates from leveraging negative data. RL (GRPO) has this term, but RFT does **not**. This highlights that negative data is the critical part of our design.
>
> Since $w(q)$ can be absorbed into the coefficient function $f^+$, we alter $w(q)$ mainly to find a function that best aligns the NFT coefficient $f(\cdot)$ with GRPO $g(\cdot)$. **Prompt weighting is a useful trick, but NOT the core algorithmic design.**
>
>
>
>
> **Q6: This paper does not clearly demonstrate that this gap is closed by self-improvement from mistakes in negative data.**
>
> **A6:** We do not really agree with the reviewer. Part of the concerns may be addressed by previous responses.  The core of our research is to demonstrate how incorporating negative data can improve traditional finetuning methods.
>
> | | base | RFT (w/o negative) | NFT (w/ negative) |
> | :--- | :---: | :---: | :---: |
> | 7B   | 31.6 | 48.3 | 51.7 |
> | 32B  | 29.6 | 52.8 | 59.2 |
>
> 1. In Table 1 and Figure 7, we clearly show that although RFT is effective, NFT still significantly outperforms it by leveraging negative data.
> 2. In Figure 8, we compare the entropy curve of NFT against RFT and show that the additional negative data helps avoid entropy collapse.
> 3. In Section 4, we prove that the theoretical gap between finetuning methods like RFT and RL methods like GRPO can be mitigated by leveraging negative data.
>
> In summary, we demonstrate a clear advantage of using negative data over RFT methods, both experimentally and theoretically.
>
>
> **Q7: NFT offers a notable improvement (AMC23), but in others losing to RFT or RL based approaches**
>
> **A7**: The results can exhibit some randomness across different runs.
>
> **NFT outperforms RFT in 5/6 benchmarks for 7B and 6/6 benchmarks in 32B experiments. We believe this demonstrates a clear and consistent improvement.**
>
> Compared with RL, NFT achieves better results in 4/6 benchmarks for both 7B and 32B experiments. We believe this is sufficient to support our claim that "NFT substantially mitigates this gap between RFT and RL," as stated in the paper.
>
> **Q8: Couldn't higher entropy be achieved by entropy regularization? Is negative data necessary?**
>
> **A8:** We believe this is still an open problem in the field. We are not entirely certain, but would like to offer some perspective.
>
> Some existing works [4] pointed out that directly applying an entropy loss, though effective on the entropy curve itself, does not truly boost exploration. Rather, an entropy increase guided by negative samples can be more beneficial. This is because the goal is not to increase entropy arbitrarily; instead, entropy serves as an indicator of whether the algorithm can boost exploration into positive answer areas.
>
> [4] The Entropy Mechanism of Reinforcement Learning for Reasoning Language Models
>
> **Q9: Minor comments.**
>
> **A9**: We thank the reviewer for the really careful reading. We have updated the paper to be more precise, including clarifying notation, removing unnecessary background, and reformulating some equations.

---

### Author Response · Authors · 2025-12-01
**Rebuttal summary**

Dear AC,

We thank you for your additional effort in handling our paper. It is unfortunate that we do not get feedback from the reviewer.

We summarize key efforts made for rebuttal and clarify some main misunderstandings received during review. We hope these can be taken into consideration.

**Core contribution of NFT.** In the past, it was usually assumed that supervised learning could only leverage good (positive) data for training. We break this biased view, and demonstrate that simply supervised training on negative data (solely) is not only feasible, but also guarantees strict convergence to an improved policy.  Large-scale (~1000 H100 GPUs) experiments on both 7B and 32B models provide an in-depth understanding of the difference between the RL and SL paradigm. NFT bridges the fundamental gap between two fields.

In rebuttal, according to reviewers' suggestions:

1. We provide new experiments showing that NFT is practically beneficial in off-policy robustness and generalizability. NFT is already successfully applied in domains like Diffusion RL [1], outperforming GRPO methods by $25 \times$, strongly supporting its generalizability.

2. We conduct new ablations to show that negative data is absolutely helpful to performance improvement.

3. We compare with additional baselines like DPO, unlikelihood training, and explicit reward conditioning to highlight the effectiveness of NFT.

4. We provide a detailed error analysis pattern for NFT.

5. We add detailed algorithm efficiency analysis and additional evaluation on coding and science

[1] DiffusionNFT: Online Diffusion Reinforcement with Forward Process

---

### Meta-Review · Area_Chair_vpfy · 2026-01-08

**Summary:**

This paper proposes NFT, a supervised learning method that leverages both correct and incorrect generations for LLM training on math reasoning. Reviews ranged from 2 to 8, showing significant disagreement.

Reviewers appreciated the clear presentation and theoretical contributions, particularly proving equivalence between NFT and GRPO under on-policy conditions. Results show NFT matches or exceeds RL methods like GRPO and DAPO.

Main concerns: whether gains come from practical modifications versus negative data use; novelty of incorporating negative examples; limited evaluation scope (only math); unclear advantages over DAPO in typical on-policy settings.

Authors provided substantive rebuttals with new experiments demonstrating: superior off-policy robustness (stable at 16x off-policy ratio where DAPO collapses); generalization to coding and science tasks; clear contribution of negative data through ablations; memory efficiency gains. Key theoretical contribution is proving convergence guarantees when training on negative data alone.

**Reviewer Concerns:**

Addressed:
- Ablations show NFT outperforms RFT by 3.4 points (7B) and 6.4 points (32B), isolating negative data contribution
- New experiments on coding and science tasks demonstrate generalization
- Memory efficiency quantified: 59.2 GB vs GRPO's 66.2 GB
- Implicit policy advantages shown versus explicit reward conditioning and naive unlikelihood baselines

Outstanding:
- On-policy performance versus DAPO shows only marginal differences in typical use cases
- SL versus RL classification remains conceptually unclear despite authors' explanation
- One reviewer (score 2) did not engage in discussion, leaving novelty concerns unresolved

**Reviewer Scores:**

Initial scores: 4, 2, 8, 4

Reviewer 1 (4): Balanced review with concerns about implicit policy benefits and mixed results. Rebuttal addressed most issues with ablations and off-policy experiments. Likely maintains 4 or increases to 5 given their openness to acceptance.

Reviewer 2 (2): Questioned novelty and conceptual advantages. Did not engage in discussion despite AC request. Score likely remains 2 due to non-participation, though rebuttals addressed core concerns about off-policy robustness and generalizability.

Reviewer 3 (8): Most enthusiastic, praised theory and empirics. Additional experiments on coding/science and efficiency analysis provided as requested. Likely maintains 8.

Reviewer 4 (4): Concerned about novelty and SL/RL categorization. Authors clarified theoretical novelty (convergence on negative-only data) with supporting experiments. Concerns largely addressed. Likely maintains 4 or slightly increases to 5.

---

### Decision · Program_Chairs · 2026-01-26

Accept (Poster)